# Seismic evidence for failed rifting in the Ligurian Basin, Western Alpine Domain

Anke Dannowski[1], Heidrun Kopp[1,2], Ingo Grevemeyer[1], Dietrich Lange[1], Martin Thorwart[2], Jörg Bialas[1], and Martin Wollatz-Vogt[1]

[1]GEOMAR Helmholtz Centre for Ocean Research Kiel, Germany
[2]CAU, Christian-Albrechts-Universität zu Kiel, Germany

*Correspondence to*: Anke Dannowski (adannowski@geomar.de)

**Abstract.** The Ligurian Basin is located in the Mediterranean Sea to the north-west of Corsica at the transition from the western Alpine orogen to the Apennine system and was generated by the south-eastward trench retreat of the Apennines-Calabrian

subduction zone. Late Oligocene to Miocene rifting caused continental extension and subsidence, leading to the opening of the basin. Yet, it remains enigmatic if rifting caused continental break-up and seafloor spreading. To reveal its lithospheric architecture, we acquired a 130-km long seismic refraction and wide-angle reflection profile in the Ligurian Basin. The seismic line was recorded in the framework of SPP2017 4D-MB, the German component of the European AlpArray initiative, and trends in a NE-SW direction at the centre of the Ligurian Basin, roughly parallel to the French coastline.

The seismic data were recorded on the newly developed GEOLOG recorder, designed at GEOMAR, and are dominated by sedimentary refractions and show mantle Pn arrivals at offsets of up to 70 km and a very prominent wide-angle Moho reflection. The main features share several characteristics (i.e. offset range, continuity), generally associated with continental settings rather than documenting oceanic crust emplaced by seafloor spreading. Seismic tomography results are complemented by gravity data and yield a ~6-8 km thick sedimentary cover and the seismic Moho at 11-13 km depth below the sea surface.

Our study reveals that the oceanic domain does not extend as far north as previously assumed. Whether Oligocene-Miocene extension led to extreme thinned continental crust or exhumed subcontinental mantle remains enigmatic. A low grade of mantle serpentinisation indicates a high rate of syn-rift sedimentation. However, rifting failed before oceanic spreading was initiated and continental crust thickens towards the NE within the northern Ligurian Basin.

## 1 Introduction

The Ligurian Sea is situated in the north-western Mediterranean Sea at the transition from the western Alpine orogen to the Apennine system. The geodynamic setting of the area is controlled by the convergence of the African and Eurasian plates (e.g. Dercourt et al., 1986). Despite the existing large collection of seismic and other geophysical data, the present-day crustal architecture of the Ligurian Basin is still under discussion and the kinematic boundaries are poorly resolved, in particular, the continent-ocean transition (COT) along the margins as well as its termination to the north-northeast. Imaging clear fault

structures within the crust has proven challenging due to the presence of thick Messinian salt layers and due to the masking

effect of the first seafloor multiple which roughly coincides with the arrival of the reflection of the acoustic basement (Béthoux et al., 2008). Deep drilling data are lacking and the magnetic data are complex and anomalies discontinuous (Bayer et al., 1973). Based on integrated seismic and magnetic data, maps indicating the extent of the oceanic domain were created (i.e. Burrus, 1984; Gueguen et al., 1998; Rollet et al., 2002), however, no axial ridge was imaged near the centre of the basin (Rollet

et al., 2002). To explain the mismatch between the expected oceanic domain and the observed seismic signal, the crust in the north-eastern basin was interpreted to be 'atypical' oceanic crust (Mauffret et al., 1995; Chamot-Rooke et al., 1997; Contrucci et al., 2001; Rollet et al., 2002). A clear change from continental to oceanic crust was only shown for the southern area of the Ligurian Basin, in the Gulf of Lion and offshore Sardinia (Gailler et al., 2009). It is proposed that the oceanic domain is separated from the continental margins by a transitional domain characterised by a high-velocity lower crust (Fig. 1). An

overview of seismic experiments until 2002 is presented in Rollet et al. (2002). Furthermore, the area was revisited or data were re-analysed with modern seismic techniques including the CROP deep seismic profiles (Finetti et al., 2005), the TGS-NOPEC and the SARDINIA profiles (Gailler et al., 2009; Jolivet et al., 2015), as well as more recent studies along the French and Italian Riviera with the 3D seismic refraction GROSMarin project (Dessa et al., 2011) and an amphibious ambient noise study (Guerin et al., 2019).

In the frame of the LOBSTER project, we obtained a new state-of-the-art seismic refraction data (Fig. 1, red line with orange and yellow triangles). Here, we present the analysis of the seismic refraction data from the central Ligurian Basin, which is the extension of a pre-existing seismic profile (Makris et al., 1999), which we call MAKRIS (Fig. 1, black line). We aim to unravel the present-day crustal structure and its nature in the centre of the Ligurian Basin, map the depth of the crust-mantle boundary (seismic Moho), and reveal the styles of deformation during the last extensional phase. We investigate the hypothesis that

Oligocene-Miocene rifting led to either extended continental crust or exhumation of sub-continental mantle below post-rift sediments in the north-eastern Ligurian Basin.

## 2 Geological structures and geodynamics of the Ligurian Sea and the Corsica-Sardinia block

The Ligurian Sea has a width of ~150 km, reaching from the northern tip of Corsica to the Ligurian coast near the city of Sanremo. It widens towards the southwest to ~175 km between Calvi (Corsica) and Cannes. South of an imaginary line between

Ajaccio (Corsica) and Toulon, the Ligurian Sea is roughly 225 km wide and opens entirely towards the Balearic Sea. The Ligurian Basin itself is smaller with a width of 70 km, 120 km, and 170 km, respectively, along the three dashed grey lines in Figure 1 and the seafloor reaches a depth of ~2700 m. The Ligurian margin is characterised by a narrow and steep slope (10-20 km) with a few listric normal faults (Finetti et al., 2005). The Corsica slope is wider (20-50 km) and the margin is characterised by several listric faults extending over a wider area (Contrucci et al., 2001; Rollet et al., 2002).

The Ligurian Sea formed as a back-arc basin at the transition from the western Alpine orogen to the Apennine system (e.g. Doglioni et al., 1997; Faccenna et al., 1997; Réhault et al., 1984). The Alpine transition is characterised by a change in subduction polarity between the two orogens (Jolivet and Faccenna, 2000; Handy et al., 2010). The Ligurian Basin is the oldest

back-arc basin in the Western Mediterranean Sea and developed from Late Oligocene to Early Miocene (Réhault and Béthoux, 1984; Roca and Desegaulx, 1992; Fernàndez et al., 1995; Jolivet and Faccenna, 2000; Rosenbaum et al., 2002; Finetti et al.,

2005; Advokaat et al., 2014). The extension is related to the south-east trench retreat of the Apennines-Calabrian subduction zone initiated in the Oligocene (Montigny et al., 1981; Réhault and Béthoux, 1984; Vigliotti and Langenheim, 1995; Gueguen et al., 1998; Rosenbaum et al., 2002; Faccenna et al., 2001).

Rifting has initiated ~30 Ma ago at a rate of ~1 cm/yr in the NE and ~2 cm/yr in the SW (Rollet et al., 2002). The initiation is associated with magmatism on land along the western Ligurian margin (Rollet et al., 2002). At roughly 21 Ma, rifting

terminated while an anticlockwise rotation of the Corsica-Sardinia block was initiated (Rollet et al., 2002; Speranza et al., 2002). During this phase, the commencement of oceanic spreading was proposed (Pascal et al., 1993; Contrucci et al., 2001; Rollet et al., 2002; Finetti et al., 2005). These authors referred to tholeiitic volcanic edifices to solidify their interpretation and interpreted the pattern of magnetic data (Bayer et al., 1973) to be a result of two main discontinuous volcanic lineaments, sub-parallel to the basin axis related to oceanic spreading and unroofing of mantle material. The opening of the Ligurian Basin

ended ~16-15 Ma ago and was associated with a second calc-alkaline volcanic phase along the Corsican margin (Rollet et al., 2002) that is linked to the migration of the subducting lithosphere towards the E-SE. The extension of the Ligurian Basin terminated and shifted to the Tyrrhenian Sea while the Apennines-Calabrian subduction zone continued to roll back further southeast until late the Messinian, ~6 Ma (Faccenna et al., 2001; Advokaat et al., 2014). The opening rate was calculated with 7.8-10.3 mm/yr (Moeller et al., 2013). In the north of the Tyrrhenian Sea, extension led to continental crustal thinning (Moeller

et al., 2013), while further south in the centre of the Tyrrhenian Basin, the mantle was exhumed and serpentinised and intruded by Mid-Ocean-Ridge type (MOR-type and intraplate basalts) (Prada et al., 2016). Similar to the Ligurian Basin, the Tyrrhenian Sea shows distributed, non-linear magnetic anomalies (Cella et al., 2008). Anomalies often coincide with volcanic islands, seamounts or other morphological units of igneous composition. During the Ocean Drilling Project (ODP) Leg 107 at site 651, serpentinised mantle rocks were drilled forming the top of the basement (Bonatti et al., 1990).

Gueguen et al. (1998) and Rollet et al. (2002) suggest that the central Ligurian Basin is comprised of oceanic crust. These authors divided the basin into different zones of continental and oceanic domains based on seismic, magnetic and gravity data (Fig. 1): (1) atypical oceanic crust with (2) transitional zones to (3) continental crust. The location of the northeast-southwest trending continent-ocean transition is proposed to be situated in the vicinity of the volcanic Tristanites Massif (Fig. 1) (Makris et al., 1999) (yellow bar perpendicular to the MAKRIS profile in Figure 1). Based on re-analysed expanding spread profiles

(ESP), Contrucci et al. (2001) proposed a 40 km wide area of oceanic crust near the Median Seamount (Fig. 1).

## 3 Data acquisition, processing, and modelling

Data at different scales resolving the subsurface structure were acquired in the Ligurian Sea in February of 2018 during the cruise MSM71 aboard the German research vessel Maria S. Merian (Kopp et al., 2018). Active seismic refraction data were

obtained along the centre of the basin. Our NE-SW trending seismic refraction and wide-angle reflection line is situated in the prolongation of an existing refraction profile in the northern Ligurian Basin (Makris et al., 1999) (Fig. 1).

## 3.1 Data acquisition and processing

The active seismic data were simultaneously recorded on short period ocean bottom seismometers (OBS) and ocean bottom hydrophones (OBH) as well as on a short streamer (280 m long) that was towed behind the vessel at 5 m water depth.
Additionally, Parasound sediment echo sounding data were recorded along the profiles. The 127.5 km long refraction seismic profile consists of 15 OBH/OBS at a station spacing of ~ 8km (Fig. 1). A total of 1079 shots were fired by an ~89-liter (5420 inch³) G-gun array, consisting of 2 sub-arrays. Each sub-array with a cluster of 2x8.5 litres (520 inch³), followed by a cluster in the middle of 2x6.2 litres (2x380 inch³, port) and 2x4.1 litres (2x250 inch³, starboard), and the third cluster again of 2x8.5 litres for both sub-arrays. The array with a string distance of 12 m was towed at 8 m below the sea-surface and 40 m behind
the vessel. A shot interval of 60 s resulted in a shot distance of ~123 m. The guns were shot at ~190 bar providing a dominant frequency band of approximately 5-70 Hz. The location of the stations on the seafloor was determined using the symmetry of the direct water arrivals from the shots on both sides. For this purpose, the direct arrival was picked and the deviation between computed and real travel times was minimised by adjusting the OBS's position along the profile. Dislocation off-line cannot be corrected with this method. For 2D traveltime modelling, the stations were projected on to the profile. The airgun shots
were recorded using newly developed GEOLOG data loggers designed at GEOMAR. All recorders operated reliably during the deployment of 2 days with a negligible absolute clock drift between -1.03 ms and +0.72 ms. The sampling frequency was 250 Hz. The data processing included the conversion of the continuous data from GEOLOG format into the standard continuous SEG-Y format using the GEOLOG programming interface. Afterwards, the continuous SEG-Y data were converted into standard trace-based SEG-Y format (Fig. 2b). Simultaneously, the clock drift was corrected, a step important for OBS
data, since the instruments cannot be continuously synchronized via GPS during deployment as commonly done onshore. A gated Wiener multi-trace deconvolution with an autocorrelation average of 51 traces was applied to the shot gathers to compress the basic wavelet, to leave the Earth's reflectivity in the seismic trace and to remove the source signature and the hydrophone and geophone responses.

## 3.2 The GEOLOG recorder

The GEOLOG is a 32-bit seismic data logger designed to digitise data from a three-component seismometer and a hydrophone. We recorded the hydrophone output on two channels (channels 1 and 5) at two different amplification levels providing well amplified long-range records (gain=16) and preventing clipped amplitudes from short-range airgun shots (gain=1) to minimise difficulties with amplitude restoration because no gain range was implemented. The gain for the three seismometer channels 2 to 4 was set to 16, which provided good signal to noise ratios for all record offsets without clipping of amplitudes. Two
additional analogue pins can be used as general-purpose input/output (GPIO) for measuring power levels for example. 3.3 V and 5 V connectors can serve external devices. Sampling intervals between 50 Hz and 4 kHz are controlled either by an atomic

clock or by a temperature compensated clock (SEASCAN). We used an external GPS receiver for synchronization of the internal clock prior and after deployment, which was driven by the GEOLOG itself. Our seismic data were stored on two micro SD cards with a volume of 32 GB each. The recorder has been tested and proved reliable for writing speeds and SD cards of
up to 128 GB (larger capacities are possible). The low power consumption of 375 mW (average battery drain) allowed us to save batteries. We used only 8 alkaline batteries per station for our short-term deployment. Thus, using lithium batteries, long-term deployments of more than 9 months can be performed. Battery power can further be saved by a delayed start of recording up to 31 days after programming. We set the recording parameters, i.e. the number of channels, gain and sampling rate, using a graphical user interface. The recorders can be programmed through any terminal program on a Windows or Linux operating
system. The programming device was connected via RS232 using an RS232-USB adapter. A second RS232 interface can be used to drive external sensors (e.g. levelling of broadband seismometers). The GPS system used for the internal clock time synchronisation was developed together with the recorder and can operate with GPS, GLONASS, GALLILEO and QZSS enabling operation worldwide and in polar regions. Besides stable output of NMEA data (defined by the National Marine Electronics Association) and a PPS (pulse-per-second) time signal, the German DCF-77 code is also available. Moreover, the
GPS system is available to deliver time or distance based trigger with TTL output, NMEA sequence and records of time stamps on an SD card.

### 3.3 Data description and analysis

The airgun shots can be followed for offsets up to 60 km on all 15 stations (Fig. 2). In general, the sections look very similar with clear sedimentary arrivals and wide-angle Moho reflections (PmP) as well as mantle phases (Pn) at a critical distance
between 25 km and 35 km to the stations (Fig. 2a). Although phase arrivals show common features in all record sections (Fig. 2a), the characteristics of the seismic phases change slightly from south to north (Fig. 2b-2d).

As a result of decreasing water depth towards the northeast, the direct wave through water (Pw) arrive later at the southern stations than at the northern stations (Fig. 2a). Arrivals from a shallow sedimentary reflection phase (PsP) occur approximately 0.5 s to 1 s after the direct wave and result from the top of salts that become shallower towards the north (as imaged in the
multichannel seismic data in Fig. 3a). The red picks (Ps1) and the orange picks (Ps2) (Fig. 2b-2d) are interpreted as refracted phases through Plio-Quaternary and older sediments, respectively. The apparent seismic velocity of the Ps2 is very constant at ~4.3 km/s to ~4.6 km/s. The phase shows many undulations and some shadow zones (Fig. 3b) caused by the salt unit that displays intense doming and is possibly disrupted by some volcanic structures that are imaged in the MCS (Fig. 3a) and the Parasound data (Fig. 3c). This phase continues as a secondary arrival (Ps3) with a similar apparent velocity of ~4.6 km/s at the
southern stations but disappears at the northern stations. Based on the apparent velocity and forward modelling, we interpret phase Ps3 as a refracted phase through the sediments. Simultaneously, when phase Ps3 disappears from OBS208 towards the north (compare to OBS209 (Fig. 2c), where Ps3 only occurs on the southern branch), an additional refracted phase (Pg) (green picks in Fig. 2c-2d) occurs with an increasing range of offsets observed on the stations northwards. The phase has an apparent velocity of ~6.2 km/s. At an offset of about 25 km, an abrupt change in the apparent seismic velocity to ~8 km/s occurs for the

160 first arrival, as typically observed in the oceanic upper mantle. The yellow picks (Fig. 2b-2d) are refracted mantle phases (Pn) that show a similar apparent seismic velocity of ~8 km/s at the northern stations. However, the critical distance at the northern stations moves to slightly larger offsets of up to ~30 km. Furthermore, an earlier very short reflection occurs at 20-25 km offset. Pn phases at the southern stations are very weak, while the PmP is relatively strong compared to typical oceanic crust characteristics. The observed slight changes in the seismic signal are accompanied by slight changes in the free-air gravity

165 anomaly around profile KM 60 (approx. 20 km south of OBS209), as discussed below.

### 3.4 P-wave traveltime tomography modelling strategy and parameters

A preliminary seismic velocity model was build using RAYINVR (Zelt, 1999) to (1) reveal the overall structure of the profile, (2) manually assign the picked phases to certain layers, and (3) serve as starting point for the travel time tomography. Travel times were picked on the hydrophone channels using the interactive analysis tool for wide-angle seismic data PASTEUP (Fujie

170 et al., 2008). The overall quality of the hydrophone data was slightly better compared to the vertical geophone channel, however, the vertical component was used for picking to confirm and to complement the picks observed on the hydrophone channel. In addition, multiples were picked when above the noise level (because of constructive interference) and where primary waves are below the noise level (Meléndez et al., 2014). Picks of water layer multiple phases were used during the forward modelling approach to confirm the layer boundaries and seismic velocities. Thereafter, a travel time tomographic

175 inversion (tomo2D from Korenaga et al., 2000) was applied to invert the seismic P-wave velocity model and yield model uncertainties. The picks were assigned pick uncertainties ranging from 20 ms for clear near offset phases (Ps1), 30 ms for intermediate offsets (Ps2 and Pg), and up to 50-70 ms for picks at larger offset (Pn and PmP) taking into account the decreased resolution due to the increased wave length of the seismic signal and the decreased signal-noise-ratio. Subsequently, all first arrivals and the mantle reflections were inverted with a set of starting models that converged to chi² values of less than 1 within

180 5 iterations. To test the model space and its limits, starting models, ranging from velocities between 1.8 km/s and 2.5 km/s at the seafloor with different velocity gradients, and ranging from 4.5 km/s to 7.5 km/s at 12-13.5 km depth to mimic the different types of crust, were manually created using RAYINVR (Zelt, 1999). The 1D starting models were hanging below the seafloor (Fig. 4c). To carefully evaluate the resulting velocity models, we used three criteria: (1) travel times need to fit the data (Fig. 2a), (2) travel time residuals, RMS misfit and chi² had to be low (i.e. chi² ~ 1), and (3) the gravity response (calculated after a

185 velocity-density conversion after Korenaga et al., 2001) of the resulting density model must yield comparable results to the satellite gravity data. Based on this evaluation, 17 models (Fig. 4c) were chosen to generate an average model for the crustal part (Fig. 4a, above the Moho) and the standard deviation was calculated (Fig. 4b). Overall, the standard deviation in the crust down to the acoustic basement is smaller than 0.15 km/s, indicating small differences between the inverted velocity models and hence an excellent resolution. Random Gaussian noise was not added, to the travel time picks, however, during modelling

190 re-picking of phases (mainly fine adjustments to the picks) did not lead to major differences in the resulting velocity model. In a further step, the average model was edited by adding different 1D profiles with mantle velocities underneath the crust-mantle boundary (inlay in Fig. 4d). A set of 14 mantle velocity starting models was used to invert for refracted mantle phases,

while the model above the seismic Moho was overdamped. Again, an average model and the standard deviation for the mantle were calculated (Fig. 4d). Standard deviations for the mantle P-wave velocities are small (<0.1 km/s), indicating a good resolution of upper mantle velocities. Lastly, the very short reflected phases interpreted to result from the top of continental crust were calculated as a floating reflector without implementing a velocity discontinuity into the model to confirm the top of crust, i.e. the crystalline basement (CB in Fig. 4a).

## 4 Results

### 4.1 Seismic P-wave velocity distribution

In general, the average P-wave velocity along the profile (Fig. 4a) shows only minor lateral variations, mainly caused by the salt layers and the corresponding tectonic features at 4-6 km depth. The uppermost portion of the velocity model is characterised by a strong velocity gradient of ~1 s$^{-1}$ that is laterally constant. P-wave velocities increase from 2.2 km/s at the seafloor to 3.5 km/s approximately 1.3 km depth below the seafloor. We interpret this unit as Plio-Quaternary sediments mixed with the upper evaporite unit after Rollet et al. (2002), using their multi-channel seismic data profile MA24 (Fig. 1, inlay profile 6). The Plio-Quaternary sediments are imaged as horizontally layered strata in the multi-channel seismic data in Figure 3a. This high velocity-gradient layer thins towards the north, from 1.5 km to 1.2 km thickness and shows slightly slower velocities at the southern end (2.2 km/s) compared to the northern end (2.4 km/s) at the seafloor. Between ~4 km and 6 km depth, the velocities range from 3.5 km/s to 4.5 km/s, and there are areas where minor velocity inversions are observed. These low velocity units have a lateral extent of up to 10 km and a velocity contrast of up to ~0.2 km/s. We identify this section as the Messinian salt unit. From 6 km to ~10 km depth, the seismic velocities increase from ~4.5 km/s at the top to 5.7 km/s at the bottom. We interpret this section as syn-rift sediments, possibly Aquitanian according to Jolivet et al. (2015), to post-rift sediments, until Pre-Messinian. We will discuss the nature of these layers in a later section, since the observed seismic velocities also account for tilted fault blocks of stretched continental crust.

In the north-eastern half of the profile, starting roughly at profile KM 70, we determine the crystalline basement (CB) (red dashed line in Fig. 4a) at a depth of 10 km to 11.5 km below the sea surface. The basement velocities increase from 5.8 km/s to 6.6 km/s (marked with "Y" in Fig. 4a); they are interpreted, based on absolute velocities, as continental crust, thickening towards the north. The acoustic basement here is at a depth of ~10 km below the sea surface. At the opposite southern half of the profile, we could not identify the CB in the OBS data. However, a strong velocity jump occurs from 5.7 km/s to ≥7.3 km/s that we interpret as the crust-mantle boundary (Moho). The uppermost mantle is characterised by seismic velocities >7.3 km/s that increase to ~8 km/s over a depth interval of 2-3 km. The histogram (Fig. 4f) images a gap in seismic velocities between 6.6 km/s and 7.3 km/s, which suggests that no fresh oceanic crust material (gabbroic rocks) is present along the profile.

## 4.2 Gravity modelling

To constrain the crustal structure along the profile, we calculated the gravity response (Talwani et al., 1959) of the final seismic velocity model and compared it to the free-air gravity anomaly derived from satellite data (Sandwell et al., 2014). The fact that the profile is situated in the centre of the basin allows us to assume that only minor 3D side-effects occur in our 2D-modelling approach, caused by topography. The velocity-depth distribution was used to assign densities by applying different density-velocity relations. The water layer is assumed to have a density of 1.03 g/cm³. Gardeners rule, $\rho = 1.74 * Vp^{0.25}$, valid for sediments between 1.5 km/s < Vp < 6.1 km/s (Gardner et al., 1974), was used for the sedimentary layers. For crystalline (non-volcanic) rocks the relation: $\rho = 0.541 + 0.3601 * Vp$ (Christensen and Mooney, 1995) was used. A density of 3.3 g/cm³ was assigned to the mantle. In areas with reduced seismic mantle velocities, the mantle density was reduced to 3.15 g/cm³ (Carlson and Miller, 2003). The converted densities explain the observed free-air gravity anomaly for the part covered by our deployed instruments. We extended the profile further northeast over the marine part of the MAKRIS line (Fig. 1, inlay profile 4). From profile KM 127.5 northwards, we related the gross density model structure to the results of the MAKRIS line (Makris et al., 1999). However, we removed a large step of 10 km in Moho depth and replaced it by a more gradually deepening Moho, which closely follows the top of the layer of underplating in the MAKRIS line. The fit of observed and calculated gravity data reasonably well supports the interpretation of a thickening continental crust towards the northeast.

## 5 Discussion

### 5.1 Nature of the crust

The seismic velocity model along our refraction profile (Fig. 4a) shows no common features of oceanic crust. Oceanic crust typically consists of a high-velocity gradient in Layer 2 and a lower velocity gradient in Layer 3 (e.g. White et al., 1992; Grevemeyer et al., 2018; Christeson et al., 2019). The absolute seismic velocities are highly variable, however, for a gabbroic crust, velocities are typically between 6.7 km/s and 7.2 km/s (Grevemeyer et al., 2018; Christeson et al., 2019). The histogram in Figure 4f shows a gap for this range of velocities suggesting the lack of a thick gabbroic layer, and thus, the lack of typical oceanic crust. In any case, the lack of seismic velocities expected for oceanic crust does not support the occurrence of larger units of oceanic crust as observed in the Central Tyrrhenian Sea (Prada et al., 2014).

Continental crust is characterised by a low seismic velocity gradient throughout the crystalline crustal layers and shows typical velocities of ~5.8 to ~6.6 km/s (Christensen and Mooney, 1995). We observe this velocity range in the northern half of the profile, starting from profile KM 70, at a depth of 10 km to 13 km (marked with "Y" in Fig. 4a). The observed seismic velocities provide only two possible interpretations: (1) hyper-extended continental crust or (2) a laterally isolated magmatic intrusion within the sedimentary units feeding the volcanic extrusion observed in the MCS and Parasound data (Fig. 3c). Based on the

gravity model (Fig. 5), we favour the first scenario of extremely thinned continental crust, which is decreasing in thickness towards the SW and may even lead to exhumed mantle during the rifting phase in the south.

The velocity model for the southern half along our refraction profile is well constrained, however, the lower part (9 km to 11 km depth), above the Moho, shows higher uncertainty compared to the shallow, sedimentary units. The depth of the crust-mantle boundary is well constrained with an uncertainty range of ±0.25 km along the southern profile half in contrast to ±0.75 km along the northern profile half (Fig. 4b). We observe seismic velocities >5.5 km/s that we interpret as fast syn-rift sediments due to a missing crystalline basement reflection. Alternatively, the change from sediments to the crystalline basement might not be characterised by a high impedance contrast, and thus, not imaged in our refraction seismic data as a strong in amplitude reflection event (compare to Fig. 2) and is expressed in a higher uncertainty of the determined CB at the northern profile end (Fig. 4). The MCS line MA24 (Rollet et al., 2002) (Fig. 1, inlay profile 6) records the acoustic basement at ~6.5 s two-way traveltime (stwt), while the seafloor occurs at 3.6 stwt (~2.7 km below sea surface). By means of a simple time to depth conversion using an average seismic velocity of 4.2 km/s, we estimate a minimum sedimentary thickness of ~6.1 km (2.9 stwt), resulting in an acoustic basement depth of ~8.8 km, as a most shallow approximation (drawn as a red dotted line in figure 4a). This line roughly fits the 5.5 km/s isoline accounting for a standard deviation of 0.2 km/s (Fig. 4b). For the southern half of our profile, this leaves a maximum continental crustal thickness of 2-2.5 km, thickening northwards.

Based on the refraction seismic data along our profile (southern half) we are not able to distinguish between sediments with high seismic velocities and extremely thinned continental crust. However, we can give a minimum and maximum continental crustal thickness, ranging from 0 km to 2.5 km. Based on the velocity model (Fig. 4a) it is not possible to distinguish whether the crystalline basement is upper, middle, or lower continental crust. The thickening of the continental crust towards the north-east is as well supported by the gravity modelling (Fig. 5). Additionally, a thickening crustal layer supports the interpretation as continental crust, since we would expect the COT to be manifested in an abrupt change from oceanic to continental crust or oceanic crust to gradually thin out towards the NE, towards the rotational pole (Rosenbaum et al., 2002), depending on the position of the profile with respect to the proposed spreading axis.

An expanding spread profile (ESP) (Le Douaran et al., 1984; Contrucci et al., 2001) crosses the northern end of our profile (Fig. 1, inlay profile 5). There the crust-mantle boundary was defined at a depth of 13-15 km while the acoustic basement was observed at ~9 km depth. Contrucci et al. (2001) retrieved crustal velocities of 6.3 - 6.9 km/s for the basin centre, which in general, is in good agreement with our results. Based on MCS data (LISA01) (Contrucci et al., 2001) with an observed major step in the basement on the Ligurian margin, they interpreted the central basin as an oceanic domain. On the Corsica margin, this major step was not observed; however, magnetic anomalies were used to constrain the interpretation. The MCS data resolve only the sedimentary portion of the crust and yield no information on the internal structures of the crystalline basement itself. Thus, a different explanation for the major step in the basement near the Ligurian margin could be that upper-crustal blocks sit on top of continental mantle similar to the Galicia margin (Nagel and Buck, 2004). Our profile only provides information on the basin centre where the absolute velocities of Le Douaran et al. (1984) and Contrucci et al. (2001) fit

continental crust velocities and support our interpretation for the northern end of the profile where we observe mantle material beneath thinned continental crust.

## 5.2 Low degree of mantle serpentinisation

Seismic velocities of unaltered mantle are >7.8 km/s (Carlson and Miller, 2003; Grevemeyer et al., 2018). Based on the seismic P-wave velocities, we interpret the uppermost mantle to be serpentinised at a low grade, which is supported by the Pn phases that are weak in amplitude at the southern stations (Fig. 2). P-wave velocities of ~7.5 km/s in the south-western half of the profile (Fig. 4a) are in-line with up to ~20% serpentinisation (Carlson and Miller, 2003). From OBS204 to OBS207, the PmP phase is extremely high in amplitude and unusually clearly visible over a wide distance of up to 20 km (in ~10 km to ~30 km offset to the station). This area (profile KM 40-KM 60) is marked by Vp > 7.8 km/s directly underneath the basement ("Z" in Fig. 4a), possibly an area of unaltered mantle due to a left over (and possibly rotated) block of continental crust as observed in other magma-poor passive margins (Bayracki et al., 2016). The fact that the mantle is only partly serpentinised suggests that either continental crustal blocks are overlying the mantle, and thus it was not fully exposed during rifting or that syn-rift sediments (nowadays showing high P-wave velocities) may have been directly deposited on top of the mantle preventing its full serpentinisation (Perez-Gussinyé et al., 2013). Thus, structurally, the Ligurian Sea is mimicking the Atlantic non-volcanic passive margins of Iberia (Minshull et al., 2014) and Goban Spur (Bullock and Minshull, 2005). However, the fast mantle in the Ligurian Sea would support a much lower degree of serpentinisation when compared to these two regions with a mantle serpentinisation of 100% at the top basement and <25% in 5-7 km depth at the Goban Spur and >75% at the top basement and <25% in 2 km depth at the Iberia margin.

In comparison to the Central Tyrrhenian Sea, where exhumed mantle is inferred, the P-wave velocities of the upper mantle in the Ligurian Sea are faster (Fig. 4a). P-wave velocities of the upper mantle in the Magnaghi and Vavilov basins in the Tyrrhenian Sea (i.e. Domain #3 in Prada et al., 2014) are significantly lower with 4.5 km/s at the top of the mantle (Prada et al., 2016). We observe significant differences between both basins: (1) Along the southern half of our profile we observe strong PmP reflections indicating a high-velocity contrast at the crust-mantle boundary, while in Domain #3 in the Tyrrhenian Sea PmP reflections are absent. (2) The Ligurian Basin was stretched ~150 km during the ~16 million year opening phase, while the Tyrrhenian Sea was stretched ~170 km within ~5 million years prior the onset of oceanic spreading (Faccenna et al., 2001; Prada et al., 2014). Thus, the stretching rate in the Tyrrhenian Sea was higher compared to the Ligurian Sea. Further, (3) the Ligurian Basin has a thick sedimentary cover of ~6-8 km, while the Tyrrhenian Sea Domain #3 shows a sedimentary cover of ~1-2 km (Prada et al., 2014). Syn-rift sedimentation was recorded in MCS data (Fig. 1, inlay profile 3) in the Gulf of Lion (Jolivet et al., 2015). The proximity of the two basins to the continental margin during their formation might result in a different syn-rift sedimentation rate that possibly was higher in the Ligurian Sea compared to the Tyrrhenian Sea. Sediments are known to reduce the permeability and thus, the amount of water that reaches the mantle rocks, necessary for serpentinisation (Ruepke et al., 2013). Two other factors can play a role for the degree of mantle serpentinisation in the Ligurian Basin: Ruepke et al.

(2013) show in thermo-tectono-stratigraphic basin models the effects of sedimentary blanketing and low stretching factors on serpentinisation. Hence our seismic velocity model (Fig. 4a) can be well explained if mantle rocks have been partially exhumed

from continental crust, without being directly exposed to sea water due to syn-rift sedimentation. Also the interpretation of extremely thinned brittle continental crust requires syn-rift sedimentation since the stretching might open fluid pathways through the crust down to the mantle and would lead to a high degree of mantle serpentinisation (Nagel and Buck, 2004).

### 5.3 Continent-ocean transition and magmatic intrusions

The MCS line MA24 (Rollet et al., 2002) was shot along an ESP profile consisting of four measurements with a spacing of ~35 km (Le Douaran et al., 1984). The two transects are crossing our profile at the southern end (Fig. 1, inlay profile 6). The MCS data resolve sedimentary units, while the seismic velocities retrieved along the ESP profile show no absolute seismic velocities similar to oceanic crust. Both transects do not map a spreading axis. Further west along the Ligurian margin, a multichannel seismic study (Jolivet et al., 2015) and a wide-angle refraction seismic study (Gailler et al., 2009) of the Ligurian

margin (Fig. 1, inlay profiles 2a and 3), in the Gulf of Lion, show a wide continent-ocean transition zone. The travel time tomography model along the OBS profiles (Gailler et al., 2009) images a succession of three domains: (1) continental, (2) transitional, and (3) oceanic towards the basin centre, following the zonation of Rollet et al. (2002). The same succession was found for both continental margins, though, the Corsica margin's transitional zone is much narrower. The transitional domain is interpreted to consist of a mixture of continental crust, exhumed mantle, and magmatic intrusions (Gailler et al., 2009; Rollet

et al., 2002). In contrast, Jolivet et al. (2015) interpret the transitional zone as exhumed lower continental crust overlying the continental mantle which is in the distal part exhumed and serpentinised. The nature of the COT along the Gulf of Lion is still debated. For example, numerical modelling of continental rifting at the magma-poor Galicia margin showed that the lower crust is scarcely preserved or absent in the continental tip (Nagel and Buck, 2004). Our velocity model at the base of the continental crust is not well enough resolved (Fig. 4b) to distinguish between upper and lower continental crust, but we

emphasize again, that we can exclude oceanic crust based on the seismic velocity structure (Fig. 4f) and the results of gravity modelling (Fig. 5) along our seismic profile. The oceanic domain on both conjugated margins in the Gulf of Lion (Fig. 1, inlay profile 2a) and offshore Sardinia (Fig. 1, inlay profile 2b) was interpreted on the basis of a 2D P-wave model derived from travel time tomography as an anomalously thin oceanic crust (4–5 km) with the typical two-layer gradients clearly characteristic of oceanic Layers 2 and 3 (Gailler et al., 2009).

The extension process in the Ligurian Basin stopped roughly 16 Ma and was replaced by the extension and opening of the Tyrrhenian Sea as the Apennines-Calabrian subduction zone continued to roll back. The magnetic data (Bayer et al., 1973; Cella et al., 2008) in both basins show a similar anomaly distribution with discontinuous, partially isolated anomalies. Prada et al. (2014) analysed a seismic refraction profile crossing the Tyrrhenian Sea from Sardinia to Italy mainland. Similar to the Ligurian Basin, the western margin is more elongated than the eastern margin. They divide the analysed profile into 3 different

domains from Sardinia to the central basin: In domain #1 continental crust thins from 22 km to 13 km over a distance of 80

km. Domain #2 is interpreted as magmatic back-arc crust with blocks of continental crust and stretches over a distance of ~80 km on the Corsican side of the basin (Prada et al., 2014). The change from continental to magmatic crust is marked by an abrupt increase of seismic velocities to >7 km/s in the lower crust, similar to the observation of Gailler et al. (2009) on the Ligurian Basin side of Sardinia. Prada et al. (2014) interpret the seismic velocities, which are slightly lower than found in 0-7 Ma old-oceanic crust, to be a result of back-arc spreading close to the active volcanic arc. Domain #3 is interpreted to be composed of serpentinised mantle to a depth of 5-6 km with basaltic intrusions and shows a width of ~140 km. Prada et al., (2014) suggest that rifting in the Central Tyrrhenian Basin started with extension of continental crust, continued with back-arc spreading, followed by mantle exhumation. Later, the area underwent magmatic episodes with magmatic intrusions into the sedimentary layer or cropping out, forming volcanoes. These volcanoes and magmatic intrusions could be related to magnetic anomalies (Prada et al., 2016). Using the Tyrrhenian Sea as an analogy, we suggest that many of the isolated magnetic anomalies in the Ligurian Sea are caused by magmatic intrusions or extrusions manifested as volcanic edifices (Median Seamount, Tristanites Massif, Monte Doria; see Fig. 1) (Rollet et al., 2002), rather than related to a spreading axis, which was indeed not mapped in MCS data so far. However, in MCS data, intrusions of volcanic sills into younger sediments were observed (Finetti et al., 2005). At the Monte Doria Seamount, 11-12 Ma old basalts with a calc-alkaline signature were sampled by dredges and submersible dives (Rollet et al., 2002; Réhault et al., 2012). The age is clearly indicating post-rift magmatism. Further, volcanism related to the slab roll-back of the Apennines-Calabrian subduction zone was observed at the Ligurian continental margin and dated to the initiation of the rifting phase (Rollet et al., 2002). Volcanism was as well associated with the end of the opening of the Ligurian Basin and related to the trench retreat of the Apennines-Calabrian subduction zone (Rollet et al., 2002). This implies that volcanism also occurred during the rifting phase and could add to the discontinuous magnetic anomalies.

## 5.4 Opening of the Ligurian Basin

The opening of the Ligurian Basin in a back-arc position during late Oligocene and early Miocene was driven by the south-east retreating Apennines-Calabria-Maghrebides subduction zone (e.g. Doglioni et al., 1997; Faccenna et al., 1997; Réhault et al., 1984; Carminati et al., 1998). The shift of active expansion from the Ligurian Basin to the Tyrrhenian Sea is considered a result of the Alpine collision that locked the Corsica-Sardinia drift towards the east and slab break-offs along the northern African margin and along the Apennines (Carminati et al., 1998). Thus, the opening of the Ligurian Basin was limited in time and space. Two different conceptual scenarios of rifting could explain our observations: (1) Rifting causing continental crust to thin until continental lower crust and mantle are exhumed and afterwards oceanic spreading is induced as observed in the Gulf of Lion (Gailler et al., 2009; Jolivet et al., 2015). (2) Rifting causing continental crust to thin until back-arc spreading is initiated and the continuation of extension leads to exhumation of mantle with magmatic intrusions (Prada et al., 2016). Depending on the scenario, our profile is situated in the Ligurian transitional domain #2 or in the Tyrrhenian domain #3. Rifting scenario (2) would imply that well developed oceanic back-arc crust should occur southeast and northwest of the

profile. The transect reaches into the area of a 3D seismic study of the Ligurian margin offshore Sanremo (Dessa et al., 2011).

The authors state that they were surprised not to see a distinct change in the velocity field at the COT. Dessa et al. (2011) could not find clear evidence for a kind of back-arc crust as shown by Prada et al. (2014) or Gailler et al. (2009). However, continental crustal thinning is well imaged. Considering these aspects, we rather favour rifting scenario (1) which is also supported by the conceptual model described by Decarlis et al. (2017) for the evolution of magma-poor rifted margins. The model includes three phases of extension: (1) An initial stretching phase forming widely distributed half-grabens in the upper crust. Afterwards

(2) a thinning phase leads to hyper-extended crust and is followed by (3) an exhumation phase during which subcontinental mantle rocks were exhumed. Yet, it is debated if we observe the latter phase in the central Ligurian Basin.

Furthermore, the Ligurian Basin width in our study area (70 - 120 km) is much narrower than further south (~200 km) where domain #2 is cumulating to ~100 km in length for both conjugated margins together, which would entirely fill the basin in our study area, leaving little or no space for oceanic spreading. This is i.e. supported by petrological and geophysical observations

at the West Iberia margin, that suggest that a COT zone can reach a width of up to 200 km (Pérez-Gussinyé, 2013).

Additionally, the opening rate becomes lower towards the north and the amount of stretching becomes less, which is probably caused by the anti-clockwise rotation of the Corsica-Sardinia block. Stretching of the crust as a result of the opening of the basin becomes less intense towards the north and thus controls the NE termination of the ultra-thin continental crust. Further, the extension of the basin decreases towards the north and assuming oceanic crust to be present, the crust should become less

thick towards the proposed ridge axis tip. However, our seismic data and gravity data indicate a gradual thickening of the crystalline crust, at least a gradual deepening of the mantle, indicating thickening continental crust northwards. This is as well supported by the magnetic data (Bayer et al., 1973), which do not show the typical oceanic crust pattern of magnetisation stripes, but rather a lateral patchy pattern of magmatic domains, supporting again the lack of oceanic spreading during the formation of the Ligurian basin in the Oligocene-Miocene. Continuing further north of our seismic line, extension led to

thinned continental crust, but the amount of extension was too small to extremely thin out the continental crust and exhume mantle.

## 6 Conclusion

The P-wave velocity model determined in this study images the uppermost lithospheric structure of the central Ligurian Basin.

Syn- and post-rift sediments of ~6-8 km thickness filled the basin during and after the 15 Ma long lasting opening phase. Based on the image of the seismic velocity distribution along the southern half of the profile it remains enigmatic if the mantle is overlain directly by syn-rift sediments or by extremely thin continental crust of up to 2.5 km. The degree of mantle serpentinisation with up to 20% is low. The northern half of the profile indicates a northward thickening of continental crust and a deepening crust-mantle boundary from 11 km to 13 km. Based on the retrieved velocity distribution, gravity modelling

and results of surrounding studies, we conclude that the extension of the Ligurian Basin led to:

(1) Extended and very thin continental crust or exhumed, partially serpentinised mantle. It remains enigmatic if the mantle is overlain directly by sediments or by extremely thinned continental crust of up to 2.5 km thickness.

(2) Continental crustal thinning from north to south related to the increase of extension with increasing distance from the rotation pole of the anti-clockwise rotation of the Corsica-Sardinia block.

Furthermore, our study documents that:

(3) Seafloor spreading and formation of mantle-derived oceanic crust was not initiated during the extension of the Ligurian Basin.

Thus, we conclude that the oceanic domain does not extend as far north as previously stated and that the transition from the continental domain and the real oceanic domain with a potential spreading axis is situated south or south-west of our seismic

line.

*Data availability.* Seismic data are available on request from the first or second author and are available via the German marine data archive PANGAEA at https://doi.org/10.1594/PANGAEA.910561.

*Competing interests.* The authors declare that they have no conflict of interest.

*Acknowledgements.* This project is funded by Deutsche Forschungsgemeinschaft (DFG), grant number KO_2961/6-1. We thank captain and crew of RV Maria S. Merian cruise MSM71 for their support during the OBS work. We thank the MSM71 cruise participants for their effort. The LOBSTER project comprises the offshore component of the AlpArray seismic network
(Hetényi et al., 2018) and is part of the German priority program SPP2017 4D-MB. Generic Mapping Tools (Wessel & Smith, 1998) were used to produce the figures. We thank the reviewers L. Jolivet, J-X. Dessa, and M. Prada for their fruitful comments.

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

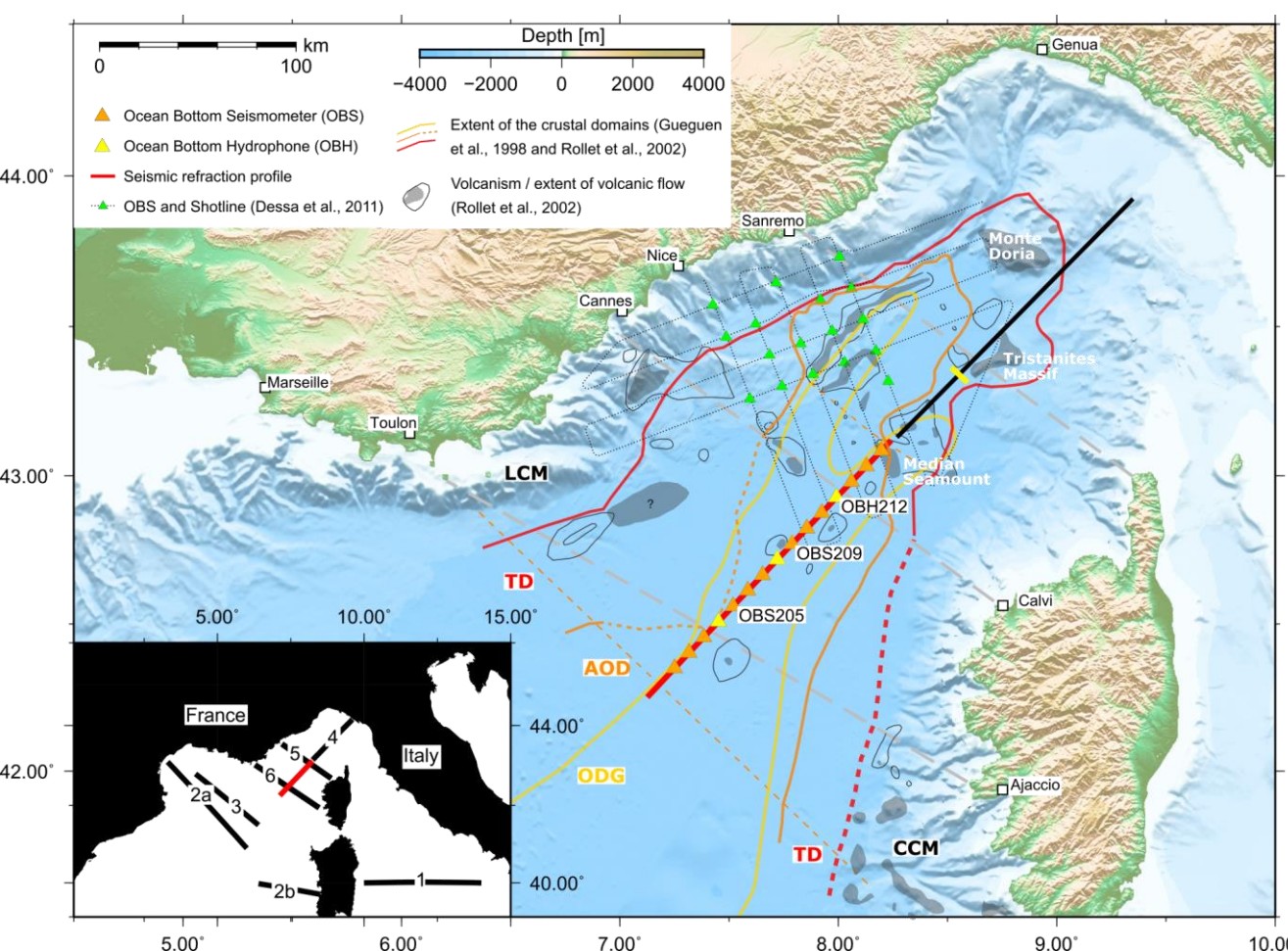

Figure 1: Relief map (GMRT data, Ryan et al., 2009) of the study area with the seismic refraction line (thick red line) and OBH/OBS locations (yellow and orange triangles, respectively) that extend the MAKRIS profile (thick long black line) (Makris et al., 1999). Thin black polygones and grey shaded areas mark volcanic extrusion after Rollet et al. (2002). The different crustal domains (Rollet et al., 2002) are marked by thin orange and red lines and are labelled with: AOD - atypical oceanic domain, CCM - Corsica continental margin, LCM - Ligurian continental margin, TD - transitional domain. A thin yellow line marks the oceanic domain (ODG) after Gueguen et al. (1998). Thin red dashed lines show proposed fracture zones (Rollet et al., 2002). Short thick yellow bar perpendicular to the MAKRIS profile marks the continent-ocean transition (COT) (Makris et al., 1999). Green triangles and thin dotted black lines are the OBS locations and shot profiles of Dessa et al. (2011). The black and white inset in the lower left corner show previous seismic refraction and reflection lines: 1 - Prada et al. (2014), 2a/2b - Gailler et al. (2009), 3 - Jolivet et al. (2015), 4 - Makris et al. (1999), 5 - Contrucci et al. (2001), 6 - MA24 from Rollet et al. (2002).

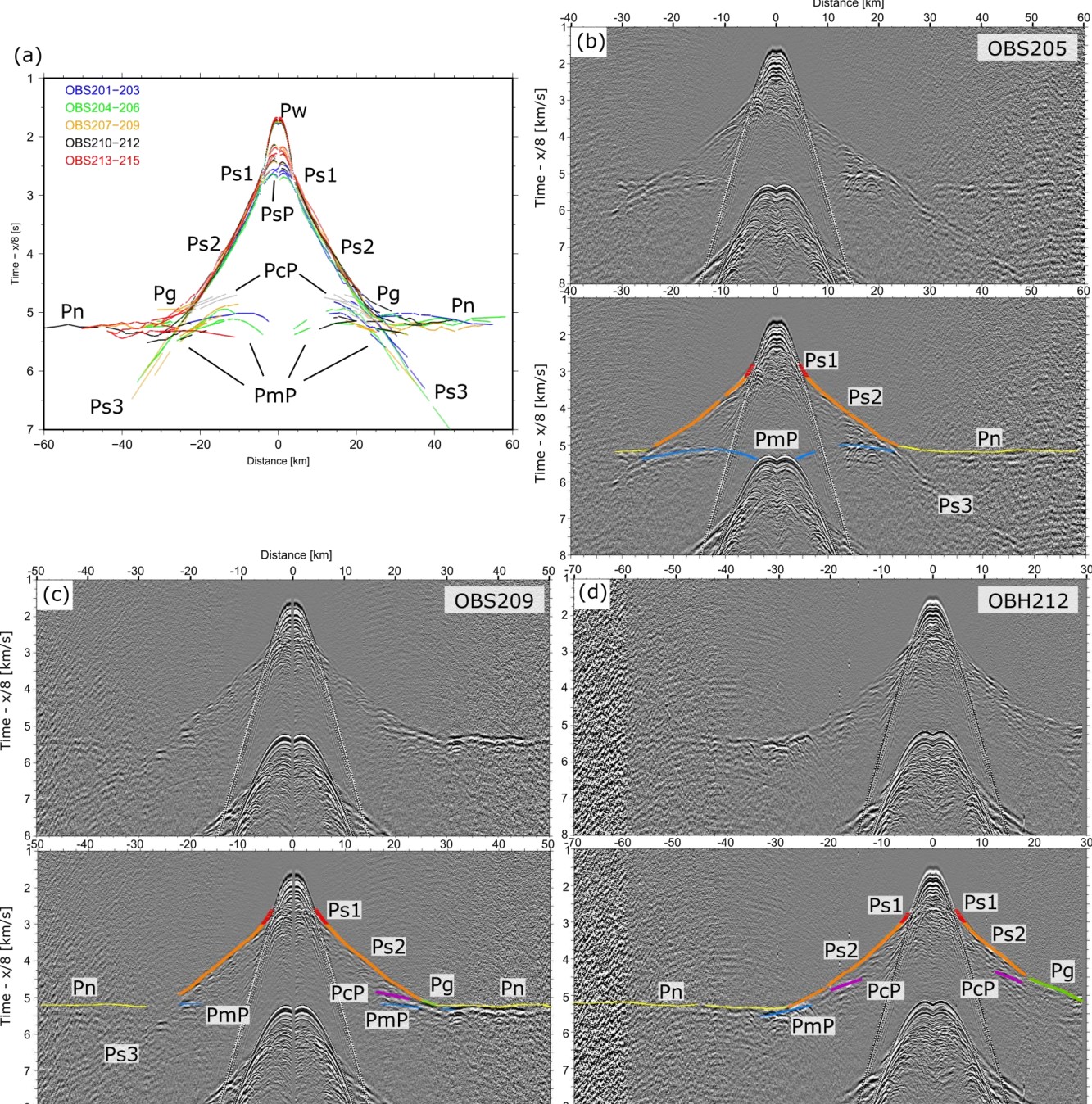

**Figure 2: (a) Stacked travel time picks of all 15 stations showing very similar arrivals suggesting an almost 1D structure along the profile. (b) Record section of station OBS205 (time reduced with a velocity of 8 km/s). The lower panel shows the calculated travel time picks from the final velocity model superimposed on the seismic data. (c) Record section and calculated travel times of station OBS209 (d) Record section and calculated travel times of station OBH212.**

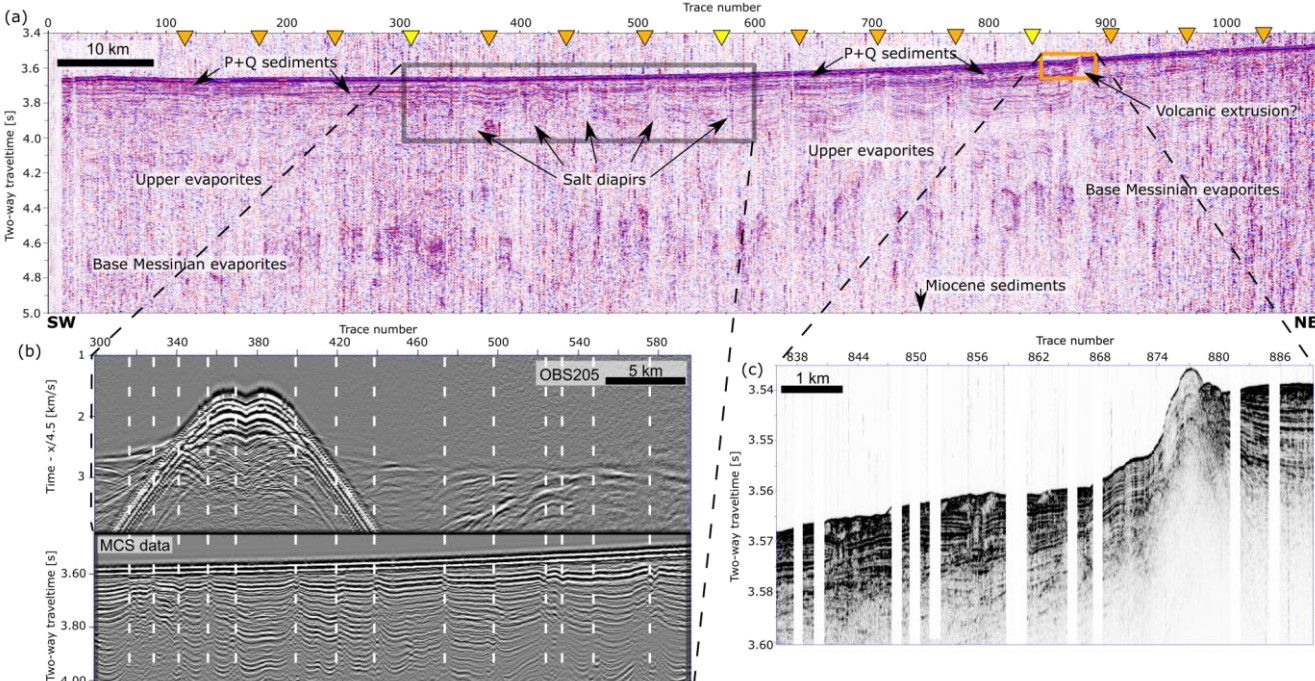

**Figure 3: (a) Multi-channel seismic data (MCS data) simultaneously shot with the refraction seismic line. The orange/yellow triangles mark the OBS/OBH positions along the profile. (b) Upper panel shows OBS205 from shot point 300 to shot point 600 with a reduction velocity of 4.5 km/s. The lower panel is a zoom into the MCS section (black box in a). The white lines show that the undulations in the sedimentary phases fit well with faults and salt diapers. (c) Parasound sediment echo sounder data (orange box in 3a).**

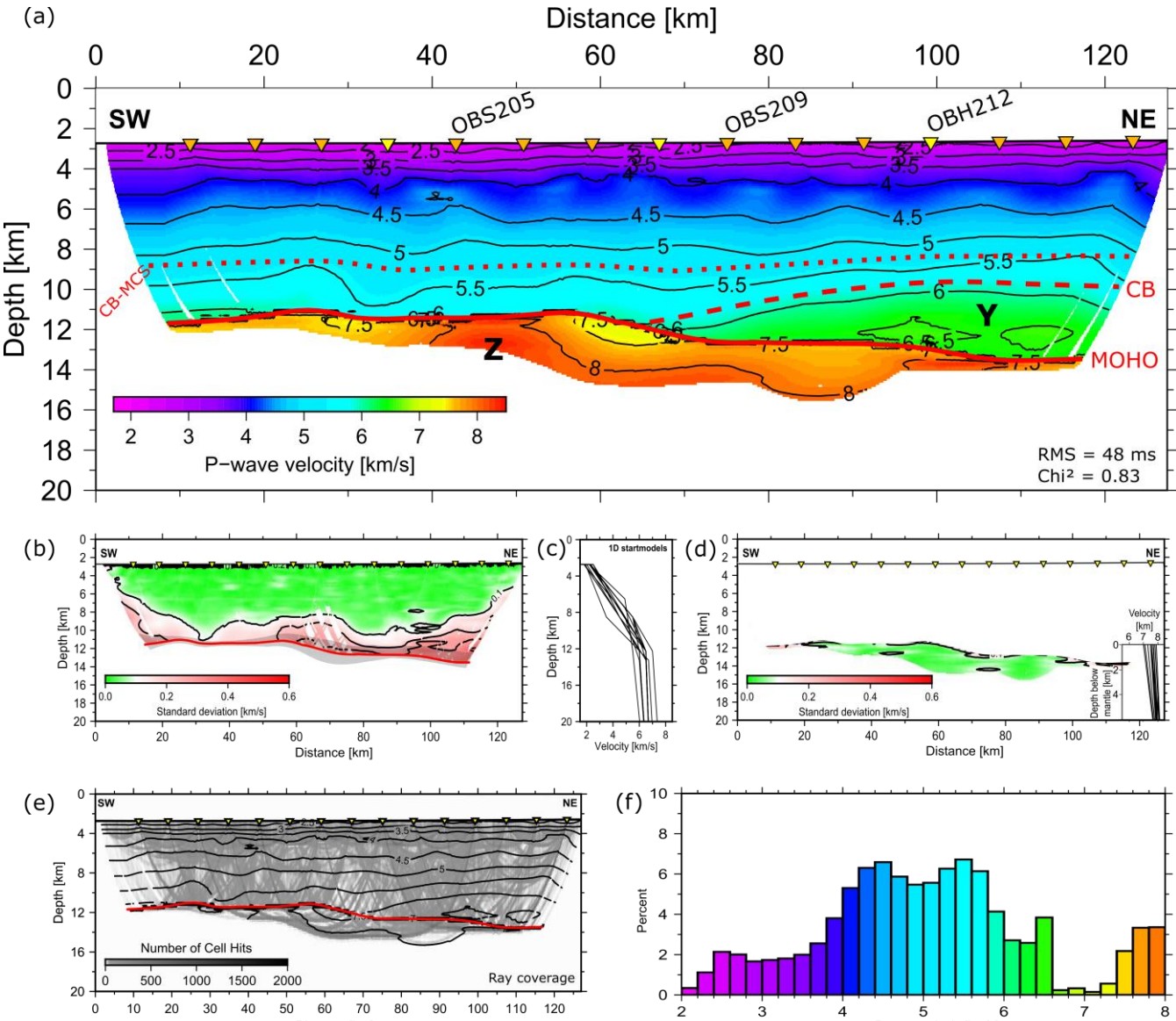

**Figure 4: (a)** Final velocity model based on the averaged velocities from the plausible starting models. The red dashed line marks the crystalline basement (CB) as determined from the refraction seismic data. The red dotted line presents the CB inferred from MCS data (CB-MCS) crossing our profile (details given in the text). The solid red line marks the crust-mantle boundary (Moho); **(b)** Standard deviation for 17 inverted velocity models, covering the crustal part down to the Moho; **(c)** Starting models used in the inversion and to calculate the resulting average model in 4a. **(d)** Standard deviation for 14 inverted velocity models (starting models in the inlay), covering the upper mantle up to the Moho; **(e)** Ray coverage for the final average velocity model; **(f)** Histogram with the velocity distribution of the final average velocity model.

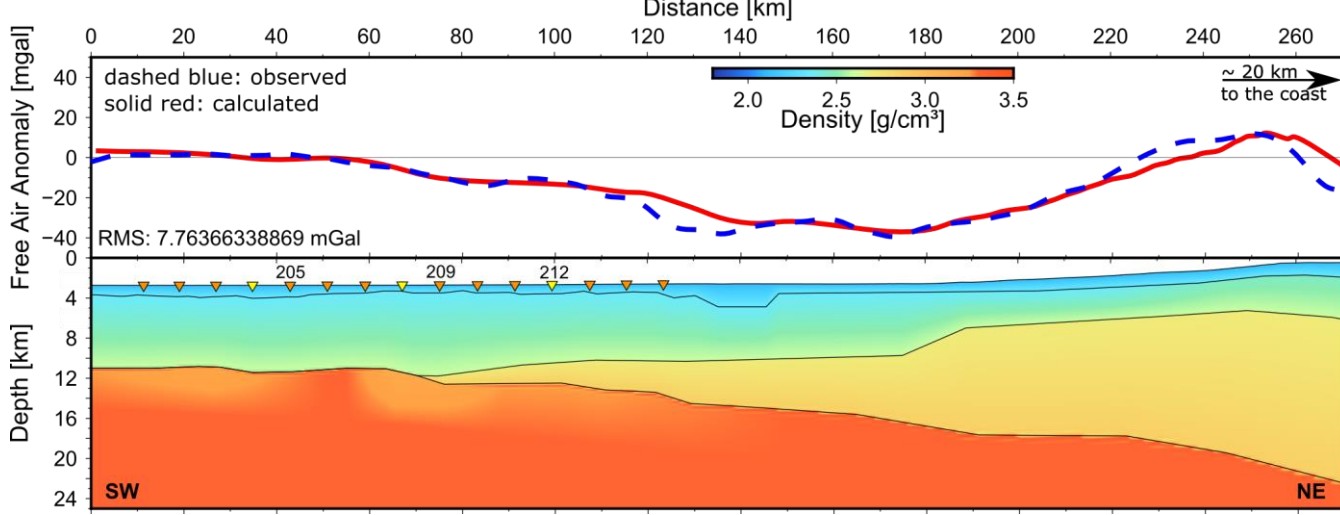

**Figure 5: Density model (lower panel) converted from seismic velocities to densities (details given in the text) for the SW part covered by seismic stations. The profile was extended towards the NE using the marine part of the seismic refraction line of Makris et al. (1999). The upper panel shows the data fit between the satellite derived free-air anomaly data (Sandwell et al., 2014 ) (dashed blue) and the model response (solid red line).**