# Peer review of "Seismic evidence for failed rifting in the Ligurian Basin, Western Alpine Domain"

_Solid Earth, 2019_

## Short Comment (SC1) · 26 Dec 2019

This paper is a welcome complement to the current knowledge of the nature of the crust in the Mediterranean back-arc basins. A debate has been active for many years on the nature of the crust and the existence or not of true oceanic crust. Recent investigations in the Tyrrhenian Sea suggest that a large part of what was previously interpreted as oceanic crust would in fact be exhumed continental mantle. This new paper addresses this question on the example of the Ligurian Basin, the north-easternmost part of the Liguro-Provençal Basin. If oceanic crust was so far supposed to be present in the center of the basin, it was always described a atypical with large volcanic intrusions

instead of a well-organized mid-ocean ridge. This new contribution shows clearly that along the entire profile, no oceanic crust is present and that exhumed mantle is found in the southern part. Whether there is or not true oceanic crust further south will remain debated but, at least for the Ligurian portion, the debate should now be closed. The paper is well written and easy to read. Although I am not a specialist of seismics I could understand the methodology and the discussion and the whole makes a convincing manuscript. The only suggestion I have to address to the authors is to open the discussion to the more general point of back-arc rifting in the Mediterranean context. The findings described here have important consequences in terms of rifting dynamics. Why does true oceanic crust does not emplace in this sort of back-arc environment is an important question.

Laurent Jolivet

---

## Referee Comment (RC1) · Jean-Xavier Dessa (Referee) · 7 Feb 2020

**1  General comments**

The manuscript presents us with a study of crustal structures over a profile crossing along strike in the central Ligurian Basin, by means of seismic refraction and seismic reflection. The main finding of the paper is that a transition from thinned continental crust to unroofed mantle covered by thick sediments is observed from northeast to southwest along the profile. In other words, no mantle-derived oceanic crust seems to be observed in the "oceanic" part of the basin. On these grounds alone, the paper

represents a very interesting contribution and is definitely worthy of publication. I have some comments on the manuscript, that can be ranked as minor but that are numerous though. I think addressing them thoroughly would help improve clarity and consistency.

**2 Specific comments and suggested corrections**

- Line 60: Dessa et al., 2011 is not a relevant citation as far as the Corsican margin is concerned. Rollet et al.'s paper and perhaps a few others should rather be cited here.

- Lines 102-103: Some more details would be welcome on the seismic source and its tuning. How many airguns? What minimum and maximum volumes? What depth of immersion? What frequency range? Considering the level of technical details provided on the GEOLOG data logger for instance, there is room for a bit more information here.

- Line 104: Still on the technical side, we are told that the symmetry of the direct arrival was used as a criterion to refine station locations. It makes perfect sense but it would be interesting to explain how this is done practically (i.e., how the position is updated from an observed asymmetry). Either a few words or a reference could be provided.

- Line 120: Technical again, and I might be unfamiliar with some recent developments, but I fail to see how an atomic clock may control the sampling rate of an autonomous sea-bottom instrument. Should we understand that in other contexts (such as in a lab...), the data logger would have this capability to be fitted with an atomic clock?

- Line 142: Nitpicking a bit, but Figure 3 has no a) and b) panels and yet, a reference is made here to Fig. 3a (although the sense of it is clear).

- Line 149: The meaning of a "longer" Pg phase is unclear to me. Should we understand that it is observed along a larger range of offsets? It might be rephrased.

- Line 153-155: The text is a bit confused here. 1) Fig. 2b is invoked but does not correspond to a northern station, as the sentence seems to imply; 2) It is not clear to me that the critical distance is larger to the north (this may have a lot to do with the fact that labels along axes in Figure 2 are not readable, see my specific comment on this point below); 3) OBS208 is not labelled in any figure and the discussed change in gravimetric data is not even located below, but 20 km south if the text is to be believed. Why then not giving the actual location of said change in km along the profile rather than with respect to a remote OBS? On a sidenote, to the best of my understanding, this location would correspond to a distance of 50 km along the profile, where I was not able to identify any change in the free air anomaly. Are we talking about the decrease observed from 60 km onward? Confusing indeed. . .

- Line 154: Same remark as for Line 142. No a) and b) panels in Figure 5. Note that this figure is referred to in the text before any reference is made to Figure 4. Normally, figures are numbered in the same order as that in which they are called in the text.

- Line 159: We are told that picking was made on hydrophone data rather than geophone. Is there a justification to this? One would arguably expect a better sensitivity of geophones. Is there any issue with data quality on geophones?

- Line 161: I do not find it completely obvious that using multiples yields more information to confirm layer boundaries. Could the authors provide a citation or a bit of explanation to support this claim?

- Line 166: Some more details explaining how the set of additional starting models were generated would be most welcome. On what assumptions were they built?

Do they span a large area in the model space? Etc. This might even warrant a figure if it can be made synthetic enough.

- Line 167: The use of $\chi^2$ criterion begs the question of travel time uncertainties and how they were assessed. No information is given about that and that too would be most welcome. A $\chi^2$ criterion of 1 only bears relevance if there is a rigorous and objective way to estimate uncertainties.

- Line 171 and 175: Standard deviation values are provided in $s$ here, which does not make any sense ito me and is not coherent with Figure 4b an 4c, where they are given in $km.s^{-1}$, as one would expect for a velocity model.

- Line 181-182: To back up the claim that lateral velocity variations are a consequence of the irregularity of the salt layer, it would be interesting to compare the wavelengths of these anomalies with those of the salt unit as imaged in MCS data. This would furthermore provide some added value to the MCS data which are practically of no use in the discussion of structures as it stands (let alone parasound data)—this point is discussed below (comment on Fig. 3).

- Line 199: Error of reference: the histogram is not in Figure 5a (which does not exist), but in Figure 4e.

- Line 217: The title of the section is wrong. "Discussion" rather than "Introduction" should feature here. The section title is likely inherited from the manuscript template...

- Line 224: Same figure, same problem as in Line 199. Figure 5e instead of 4e.

- Line 267-268: It is not clear to me what the authors mean with a less evolved crust. Do they mean the thickness of it? Its nature? As a result, I find the meaning of the last two sentences of this paragraph rather enigmatic.

- Line 309: I find the sentence about the preference of the authors for a magmatic origin to the observed magnetic anomalies rather than a relation with an unmapped spreading axis to be quite an understatement as their own results seem to completely rule out the possibility of a ridge axis (no oceanic crust is an overwhelming argument against the existence of any accretion axis at any time here I believe). More generally, I think this very interesting result and its implications are not highlighted enough in the discussion.

- Fig. 1: A lot of features on the map, some not very visible, some practically invisible due to a poor choice of non-contrasting colors with respect to the bathymetric background or to the use of thin lines. Rollet et al. refer to "atypical oceanic domain" instead of "atypical oceanic crust". Since the main result of this study is to rule out the existence of an oceanic igneous crust, the reference to an "Atypical Oceanic Crust" is a bit confusing here. I would suggest the same appellation as in Rollet et al.

- Fig. 2: As mentioned above, labels along axes should be made readable for all data plots.

- Fig. 3: Features in the MCS profile are barely visible and poorly discussed in the text. This observation holds even more true for parasound results which do not back results at all. I think dropping them could be considered.

---

## Referee Comment (RC2) · Manel Prada (Referee) · 14 Feb 2020

The manuscript "Oligocene-Miocene extension led to mantle exhumation in the central Ligurian Basin, Western Alpine Domain" by A. Dannowski and co-authors present new constraints on the petrological nature of the basement in the Ligurian Basin from new wide-angle seismic data and travel-time tomography. The authors show that rather than oceanic crust, as previously interpreted in the area, the northwestern region of the basin experienced crustal thinning and later mantle exhumation.

However, I found the occurrence of mantle exhumation difficult to reconcile with the velocity structure of the uppermost mantle presented here. Considering that the mantle

is fully exposed to the seawater during the opening of the basins, I found strange that the top of the mantle does not show the typical velocity gradient of exhumed mantle regions, in which Vp increases progressively from ∼4.6 km/s (100 % alteration) to 7.8-8.0 km/s (no alteration) (Minshull 2009; https://doi.org/10.1016/j.crte.2008.09.003; Prada et al., 2015 doi: 10.1093/gji/ggv271). While I agree that there is no oceanic crust, the lack of an exhumed mantle-like Vp vertical gradient implies that the mantle was not fully hydrated and thus, exhumed, possibly because of the presence of syn-rift sediments or the existence of hyperextended continental crust. In fact, this interpretation fits nicely with the model. Lower crustal velocities are > 5.5 km/s, which may well be indicative of tilted fault blocks and rotated syn-rift sediments (e.g. Bayracki et al., 2016, Nature Geoscience, DOI: 10.1038/NGEO2671). The top of the continental basement in these settings can be really rough, and thus difficult to identify in OBS data. The fact that you don't see it, doesn't mean it's not there. In addition, mantle Vp is close to 8 km/s in some regions (e.g. beneath OBS205), while it decreases in others to < 7.5 km/s. This pattern resembles the mantle structure underlying continental tilted blocks reported in other rifted margins such as Galicia (Bayracki et al., 2016) and the Porcupine Basin (Prada et al., 2017 EPSL; http://dx.doi.org/10.1016/j.epsl.2017.06.040). Such pattern is attributed to the fault-controlled water influx to the mantle during rifting (Bayracki et al., 2016). In light of these observations, I advise the authors to reconsider their interpretation. Apart from this aspect, I also found some issues during the modelling and in section 5.4 that, if tackled, can help to improve the robustness of the final model, and thus, strengthen the paper. I discuss them the bellow.

Regardless of these issues, the paper unequivocally demonstrates that there is no oceanic crust in this region of the Ligurian Basin, and that is of great relevance for the community working on the Mediterranean region. This study fits nicely with the goals of Solid Earth, and thus, I strongly encourage the authors to tackle all these aspects and resubmit the manuscript for its publication. Best regards, Manel Prada

Major issues: The authors use forward modelling, I presume, to explore the lateral

consistency of the seismic phases observed in each receiver. Then, they use this preliminary model as input for the tomography. However, it is confusing the way the authors describe and apply the layer stripping strategy. The authors say, "In a first step only near offset picks with distances smaller than 15 km were inverted." This is rather confusing. It seems that the authors have inverted the travel times within 15 km of offset from each receiver, independently of the seismic phase they correspond to. It would be better to explicitly mention the type of seismic phases that the authors have included in the first step, which I guess by Figure 4, are all sedimentary and crustal phases, plus PmP. One would also appreciate more details on the layer stripping strategy. Did the authors overdamped the result of the crustal layer when inverting for mantle phases? On the other hand, the authors follow some sort of Monte Carlo analysis to assess the space of possible solutions but they only use 17 models for the crustal level and even a lower number for mantle phases, 12. The final standard deviation is low in Fig. 4. My concern is that given the low number of realizations tested the initial standard deviation (which one would appreciate seeing in the supplementary material) might below as well. I suggest testing at least 100 models for each layer, which is what is commonly done in this type of study to assess the uncertainty of model parameters. The outcome of this uncertainty analysis in its present form is not convincing which may lead to skepticism of the final interpretation. In addition, the authors may want to provide more details on this type of statistical test, right now is a bit vague. How are the initial models? Are they randomly created or they are derived from the forward modelling? The authors could add figures of the initial models, initial standard deviation, as well as the results from forward modelling in the supplementary material. Do they add Gaussian random noise to the picks (I would encourage them to add this to the test)? The gravity modelling could be also improved as well. The authors could show how the gravity response derived from a density model with a homogeneous mantle density of 3.3 g/cc compares with the model they have and the observed anomaly. That would help to discern between serpentinized mantle and non-altered mantle rock, which in turn would allow to strengthen the hypothesis of the paper. Line 350-351 and all section

5.4: "seafloor spreading and formation of oceanic crust was not initiated during the extension of the Ligurian Basin.". I would be more cautious here, it seems that the authors are saying that there is no oceanic crust in the whole Ligurian basin. Extension in this basin increases from north to south and as in the Tyrrhenian formation processes may significantly change from the north (region imaged in this study) to the south.

Minor changes: Line 18: augmented -> complemented

Line 22-23: "exhumation of sub-continental mantle which eventually became serpentinised". According to the models of mantle exhumation crustal faulting initiates the hydration of the mantle during rifting. Thus, serpentinization occurs before the exhumation. The authors should modify this sentence accordingly

Section 3.2 The GEOLOG recorder. This section is a bit out of place since this is not a technical paper and thus, it distracts the reader from the main point. I suggest moving this section to supplementary material and briefly mentioning the GEOLOG recording system in section 3.1.

Line 154: The gravimetric data (Fig. 5a) show a change approx. 20 km south of OBS208. Please add the numbering of OBS in Figure 5a

Figure 1: There is a bracket missing in Rollet et al. (2002), and it would be good to see the numbering of the OBS/H shown in Fig. 2 instead of OBS/H 201-208-215

---

## Author Comment (AC1) · 28 Feb 2020

**Reviewer #1 (Laurent Jolivet)**

This paper is a welcome complement to the current knowledge of the nature of the crust in the Mediterranean back-arc basins. A debate has been active for many years on the nature of the crust and the existence or not of true oceanic crust. Recent investigations in the Tyrrhenian Sea suggest that a large part of what was previously interpreted as oceanic crust would in fact be exhumed continental mantle. This new paper addresses this question on the example of the Ligurian Basin, the north-easternmost part of the Liguro-Provençal Basin. If oceanic crust was so far supposed to be present in the center of the basin, it was always described a atypical with large volcanic intrusions instead of a well-organized mid-ocean ridge. This new contribution shows clearly that along the entire profile, no oceanic crust is present and that exhumed mantle is found in the southern part. Whether there is or not true oceanic crust further south will remain debated but, at least for the Ligurian portion, the debate should now be closed. The paper is well written and easy to read. Although I am not a specialist of seismics I could understand the methodology and the discussion and the whole makes a convincing manuscript.

The only suggestion I have to address to the authors is to open the discussion to the more general point of back-arc rifting in the Mediterranean context. The findings described here have important consequences in terms of rifting dynamics. Why does true oceanic crust does not emplace in this sort of back-arc environment is an important question.

Indeed, a discussion on the back-arc rifting in the broader Mediterranean context would be interesting and worthwhile. We feel that we can provide only limited input on the raised question regarding the lack of oceanic crust in other back-arc environments in the Mediterranean solely based on the seismic profile P02 presented here. In fact, it is not known whether other back-arc environments show a similar structure as the Ligurian Basin. However, the presented new findings for the Ligurian Basin are of general interest in themselves and merit a presentation in a focussed way.
We added a more general introducing section to chapter 5.4 to show possible driving limits to the opening of the Ligurian back-arc basin:
*"The opening of the Ligurian Basin in a back-arc position during late Oligocene and early Miocene that was driven by the south-east retreating Apennines-Calabria-Maghrebides subduction zone (e.g. Doglioni et al., 1997; Faccenna et al., 1997; Carminati et al. 1998; Rehault et al., 1984). The shift of active expansion from the Ligurian basin to the Tyrrhenian Sea is considered as a result of the Alpine collision that locked the Corsica-Sardinia drift towards the east and slab break-offs along the northern African margin and along the Apennines (Carminati et al., 1998). Thus, the opening of the Ligurian Basin was limited in time and space."*

---

## Author Comment (AC2) · 28 Feb 2020

**Reviewer #2 (Jean-Xavier Dessa)**

**1 General comments**

The manuscript presents us with a study of crustal structures over a profile crossing along strike in the central Ligurian Basin, by means of seismic refraction and seismic reflection. The main finding of the paper is that a transition from thinned continental crust to unroofed mantle covered by thick sediments is observed from northeast to southwest along the profile. In other words, no mantle-derived oceanic crust seems to be observed in the "oceanic" part of the basin. On these grounds alone, the paper represents a very interesting contribution and is definitely worthy of publication. I have some comments on the manuscript, that can be ranked as minor but that are numerous though. I think addressing them thoroughly would help improve clarity and consistency.

2 Specific comments and suggested corrections

The line numbering in the word template is different to the created PDF, however, we think that we easily could find the sections pointed out.

• Line 60: Dessa et al., 2011 is not a relevant citation as far as the Corsican margin is concerned. Rollet et al.'s paper and perhaps a few others should rather be cited here.

**Indeed, we changed the reference to Contrucci et al., 2001 and Rollet et al., 2002.**

• Lines 102-103: Some more details would be welcome on the seismic source and its tuning. How many airguns? What minimum and maximum volumes? What depth of immersion? What frequency range? Considering the level of technical details provided on the GEOLOG data logger for instance, there is room for a bit more information here.

**We included more details on the airgun system itself:**

"A total of 1079 shots were fired by an ~89-liter (5420 inch3) G-gun array, consisting of 2 sub-arrays. Each sub-array with a cluster of 2x8.5 litres (520 inch3), followed by a cluster in the middle of 2x6.2 litres (2x380 inch3, port) and 2x4.1 litres (2x250 inch3, starboard), and the third cluster again of 2x8.5 litres for both sub-arrays. The array with a string distance of 12 m was towed at 8 m below the seasurface and 40 m behind the vessel. A shot interval of 60 s resulted in a shot distance of ~123 m. The guns were shot at ~190 bar providing a dominant frequency band of approximately 5-70 Hz."

• Line 104: Still on the technical side, we are told that the symmetry of the direct arrival was used as a criterion to refine station locations. It makes perfect sense but it would be interesting to explain how this is done practically (i.e., how the position is updated from an observed asymmetry). Either a few words or a reference could be provided.

**We included more details (Lines 110-113):**

"For this purpose, the direct arrival was picked and the deviation between computed and real travel times was minimised by adjusting the OBS's position along the profile. Dislocation off-line cannot be corrected with this method. For 2D traveltime modelling, the stations were projected on to the profile."

• Line 120: Technical again, and I might be unfamiliar with some recent developments, but I fail to see how an atomic clock may control the sampling rate of an autonomous sea-bottom instrument. Should we understand that in other contexts (such as in a lab...), the data logger would have this capability to be fitted with an atomic clock?

Indeed, the internal clock used in some of the recorders is a clock, which is controlled by the oscillation frequency of atoms (i.e. caesium), giving very high accuracy, compared to quartz oscillators. On the

other hand atomic clocks consume much more power (probably double the rate) than a usual simple quartz oscillator, which is a disadvantage for long term deployments.

• Line 142: Nitpicking a bit, but Figure 3 has no a) and b) panels and yet, a reference is made here to Fig. 3a (although the sense of it is clear).

The labels "a)" and "b)" have been too small. We enlarged it for better visibility and included a panel c).

• Line 149: The meaning of a "longer" Pg phase is unclear to me. Should we understand that it is observed along a larger range of offsets? It might be rephrased.

Re-phrased: "Simultaneously, when phase Ps3 disappears (from OBS208 towards the north (compare to OBS209 (Fig. 2c), where Ps3 only occurs on the southern branch), an additional refracted phase (Pg) (green picks in Fig. 2c-2d) occurs with an increasing range of offsets observed on the stations and becomes longer northwards."

• Line 153-155: The text is a bit confused here. 1) Fig. 2b is invoked but does not correspond to a northern station, as the sentence seems to imply; 2) It is not clear to me that the critical distance is larger to the north (this may have a lot to do with the fact that labels along axes in Figure 2 are not readable, see my specific comment on this point below); 3) OBS208 is not labelled in any figure and the discussed change in gravimetric data is not even located below, but 20 km south if the text is to be believed. Why then not giving the actual location of said change in km along the profile rather than with respect to a remote OBS? On a sidenote, to the best of my understanding, this location would correspond to a distance of 50 km along the profile, where I was not able to identify any change in the free air anomaly. Are we talking about the decrease observed from 60 km onward? Confusing indeed ...

We re-phrased this section in order to make it clearer to the reader:

- 1) Re-phrased the explanation of Ps3 (yellow phase, Figure 2) and then explain the slight changes from south to north.
- 2) We increased the axis labels on Figure 2 and adjusted a little bit the position of the phase labels. The difference in critical distance between the northern and the southern stations is very small (~5-10 km) and is maybe better to see in Fig. 2a.
- 3) OBS208 was wrong! Changed to OBS209. To follow the order of Figures we removed the link to Fig. 5. It was simply to give the reader a view to the second dataset to directly follow the changes, since they are very small. We do not want to change the order of figures, since the gravity modelling is based on the seismic results.

Yes, 60 km along the profile and onward towards the north, and it should be seen from OBS209. Indeed a profile KM will make the description much clearer. Added.

• Line 154: Same remark as for Line 142. No a) and b) panels in Figure 5. Note that this figure is referred to in the text before any reference is made to Figure 4. Normally, figures are numbered in the same order as that in which they are called in the text.

Figure 5 does not really have a panel (a) and (b) since for example the colour scale of the gravity model is at the top of the figure. We modified the text so that we now call the Figures in the correct order.

• Line 159: We are told that picking was made on hydrophone data rather than geophone. Is there a justification to this? One would arguably expect a better sensitivity of geophones. Is there any issue with data quality on geophones?

The data quality of the geophones is commonly controlled by the instrument coupling to the seafloor and thus varies largely between different study areas. Sometimes the geophone shows a similar or better S/N ratio than the hydrophone on our data set, also dependent on the offset range. While single instruments show a similar quality than the hydrophone, overall the hydrophone data was more robust for all stations along the profile. We included the following sentence on the data quality:

"The overall quality of the hydrophone data was slightly better compared to the vertical geophone channel, however, the vertical component was used for picking to confirm..."

• Line 161: I do not find it completely obvious that using multiples yields more information to confirm layer boundaries. Could the authors provide a citation or a bit of explanation to support this claim?

We inserted and explained in the text and gave a citation (Meléndez et al., 2014):

"The vertical seismometer component was used for picking to confirm and to complement the picks observed on the hydrophone channel. In addition, multiples were picked when above the noise level (because of constructive interference) and where primary waves are below the noise level (Meléndez et al., 2014)."

• Line 166: Some more details explaining how the set of additional starting models were generated would be most welcome. On what assumptions were they built? Do they span a large area in the model space? Etc. This might even warrant a figure if it can be made synthetic enough.

We added some text to clarify:

"To test the model space and its limits, starting models, ranging from velocities between 1.8 km/s and 2.5 km/s at the seafloor with different velocity gradients, and ranging from 4.5 km/s to 7.5 km/s at 12-13.5 km depth to mimic the different types of crust, were manually created using RAYINVR (Zelt, 1999)."

• Line 167: The use of 2 criterion begs the question of travel time uncertainties and how they were assessed. No information is given about that and that too would be most welcome. A 2 criterion of 1 only bears relevance if there is a rigorous and objective way to estimate uncertainties.

We included a sentence on the size of pick uncertainties that were assigned to the different picked phases (Lines 178-180):

"The picks were assigned pick uncertainties ranging from 20 ms for clear near offset phases (Ps1), 30 ms for intermediate offsets (Ps2 and Pg), and up to 50-70 ms for picks at larger offset (Pn and PmP) taking into account the decreased resolution due to the increased wave length of the seismic signal and the decreased signal-noise-ratio."

• Line 171 and 175: Standard deviation values are provided in s here, which does not make any sense ito me and is not coherent with Figure 4b an 4c, where they are given in km/s, as one would expect for a velocity model.

**Corrected.**

• Line 181-182: To back up the claim that lateral velocity variations are a consequence of the irregularity of the salt layer, it would be interesting to compare the wavelengths of these anomalies with those of the salt unit as imaged in MCS data. This would furthermore provide some added value to the MCS data which are practically of no use in the discussion of structures as it stands (let alone parasound data)—this point is discussed below (comment on Fig. 3).

We inserted a third panel (Fig. 3b) which compares the MCS data with the OBS data within the range of 300 traces and linked this figure to the text.

• Line 199: Error of reference: the histogram is not in Figure 5a (which does not exist), but in Figure 4e.

**Changed.**

• Line 217: The title of the section is wrong. "Discussion" rather than "Introduction" should feature here. The section title is likely inherited from the manuscript template...

Corrected - wrong copy paste into the template.

• Line 224: Same figure, same problem as in Line 199. Figure 5e instead of 4e.

**Corrected.**

• Line 267-268: It is not clear to me what the authors mean with a less evolved crust. Do they mean the thickness of it? Its nature? As a result, I find the meaning of the last two sentences of this paragraph rather enigmatic.

Changed "less evolved" to "less thick". Added that these observations indicate thickening continental crust towards the North.

• Line 309: I find the sentence about the preference of the authors for a magmatic origin to the observed magnetic anomalies rather than a relation with an unmapped spreading axis to be quite an understatement as their own results seem to completely rule out the possibility of a ridge axis (no oceanic crust is an overwhelming argument against the existence of any accretion axis at any time here I believe). More generally, I think this very interesting result and its implications are not highlighted enough in the discussion.

We changed the title in order to focus on our main finding. We changed the abstract and the style of the conclusion to bullet points to improve the visibility of our findings and highlight them.

• Fig. 1: A lot of features on the map, some not very visible, some practically invisible due to a poor choice of non-contrasting colors with respect to the bathymetric background or to the use of thin lines. Rollet et al. refer to "atypical oceanic domain" instead of "atypical oceanic crust". Since the main result of this study is to rule out the existence of an oceanic igneous crust, the reference to an "Atypical Oceanic Crust" is a bit confusing here. I would suggest the same appellation as in Rollet et al.

Changed to atypical oceanic domain (AOD) in the figure. Reduced the contrast of the map and enlarged the contrast and thickness of the lines and objects of importance.

• Fig. 2: As mentioned above, labels along axes should be made readable for all data plots.

Axis labels enlarged and phase labels slightly adjusted.

• Fig. 3: Features in the MCS profile are barely visible and poorly discussed in the text. This observation holds even more true for parasound results which do not back results at all. I think dropping them could be considered.

We inserted a third part into the figure (Fig 3, panel b) that compares the undulations of the first arrival phase in the OBS with the MCS data. We like to keep this figure, since it provides a good impression on the complexity of the shallow portion of the subsurface and shows the entire data range acquired along this profile.

---

## Author Comment (AC3) · 28 Feb 2020

**Reviewer #3 (Manel Prada)**

1 General comments

• The manuscript "Oligocene-Miocene extension led to mantle exhumation in the central Ligurian Basin, Western Alpine Domain" by A. Dannowski and co-authors present new constraints on the petrological nature of the basement in the Ligurian Basin from new wide-angle seismic data and travel-time tomography. The authors show that rather than oceanic crust, as previously interpreted in the area, the northwestern region of the basin experienced crustal thinning and later mantle exhumation. However, I found the occurrence of mantle exhumation difficult to reconcile with the velocity structure of the uppermost mantle presented here. Considering that the mantle is fully exposed to the seawater during the opening of the basins, I found strange that the top of the mantle does not show the typical velocity gradient of exhumed mantle regions, in which Vp increases progressively from ~4.6 km/s (100 % alteration) to 7.8-8.0 km/s (no alteration) (Minshull 2009; https://doi.org/10.1016/j.crte.2008.09.003; Prada et al., 2015 doi: 10.1093/gji/ggv271). While I agree that there is no oceanic crust, the lack of an exhumed mantle-like Vp vertical gradient implies that the mantle was not fully hydrated and thus, exhumed, possibly because of the presence of syn-rift sediments or the existence of hyperextended continental crust. In fact, this interpretation fits nicely with the model. Lower crustal velocities are > 5.5 km/s, which may well be indicative of tilted fault blocks and rotated syn-rift sediments (e.g. Bayracki et al., 2016, Nature Geoscience, DOI: 10.1038/NGEO2671). The top of the continental basement in these settings can be really rough, and thus difficult to identify in OBS data. The fact that you don't see it, doesn't mean it's not there. In addition, mantle Vp is close to 8 km/s in some regions (e.g. beneath OBS205), while it decreases in others to < 7.5 km/s. This pattern resembles the mantle structure underlying continental tilted blocks reported in other rifted margins such as Galicia (Bayracki et al., 2016) and the Porcupine Basin (Prada et al., 2017 EPSL; http://dx.doi.org/10.1016/j.epsl.2017.06.040). Such pattern is attributed to the fault-controlled water influx to the mantle during rifting (Bayracki et al., 2016). In light of these observations, I advise the authors to reconsider their interpretation. Apart from this aspect, I also found some issues during the modelling and in section 5.4 that, if tackled, can help to improve the robustness of the final model, and thus, strengthen the paper. I discuss them the bellow.

We agree that the concept of mantle exhumation poses some interesting aspects and thank the reviewer for pointing these out. Indeed our data may not provide the information if sediments are underlain by thinned continental crust or exhumed mantle. We now discuss both scenarios. Exhumed mantle does not necessarily imply to be exposed directly on the seafloor as we mentioned in lines 270-275. We now additionally clarify this in the discussion by adding extra paragraphs in sections 5.1 and 5.2. We re-phrased the manuscript at several places to open up the discussion about the continental material (thinned continental crust or exhumed subcontinental mantle) underlying Ligurian Basin.

In contrast to the Tyrrhenian Sea we observe a strong in amplitude PmP reflection, which indicates a high velocity contrast at the crust-mantle boundary. The nature of the velocities >5.5 km/s can be debated and we cannot distinguish between fast sediments and left over rotated continental crust blocks (rotated the orthogonal direction to the profile). Sediments have to play a role during rifting and mantle serpentinisation (Ruepke et al., 2013), else we would observe a much higher rate of mantle serpentinisation, we agree, also in areas with remnant blocks of continental crust.

We changed the manuscript title to: *"Seismic evidence for failed rifting in the Ligurian Basin, Western Alpine Domain"*

Regardless of these issues, the paper unequivocally demonstrates that there is no oceanic crust in this region of the Ligurian Basin, and that is of great relevance for the community working on the

Mediterranean region. This study fits nicely with the goals of Solid Earth, and thus, I strongly encourage the authors to tackle all these aspects and resubmit the manuscript for its publication.
Best regards, Manel Prada
Major issues:

• The authors use forward modelling, I presume, to explore the lateral consistency of the seismic phases observed in each receiver. Then, they use this preliminary model as input for the tomography. However, it is confusing the way the authors describe and apply the layer stripping strategy. The authors say, "In a first step only near offset picks with distances smaller than 15 km were inverted." This is rather confusing. It seems that the authors have inverted the travel times within 15 km of offset from each receiver, independently of the seismic phase they correspond to. It would be better to explicitly mention the type of seismic phases that the authors have included in the first step, which I guess by Figure 4, are all sedimentary and crustal phases, plus PmP.

We removed the sentence since it is rather confusing and not necessary to explain the modelling strategy for the final average velocity model. This actually was a process that allowed us to get better acquainted with the data and the model behaviour and which will not be visible in the final results. But indeed, in first steps we only selected picks up to 15 km offset to image the uppermost sediments.

• One would also appreciate more details on the layer stripping strategy. Did the authors overdamped the result of the crustal layer when inverting for mantle phases?

We overdamped the model and included this in the text.

• On the other hand, the authors follow some sort of Monte Carlo analysis to assess the space of possible solutions but they only use 17 models for the crustal level and even a lower number for mantle phases, 12. The final standard deviation is low in Fig. 4. My concern is that given the low number of realizations tested the initial standard deviation (which one would appreciate seeing in the supplementary material) might below as well. I suggest testing at least 100 models for each layer, which is what is commonly done in this type of study to assess the uncertainty of model parameters. The outcome of this uncertainty analysis in its present form is not convincing which may lead to skepticism of the final interpretation.

In contrast to 100 or even 1000 different models, to call it Monte Carlo analysis, we preferred to calculate a smaller set of models to test a wider model space. In contrast to the commonly performed automated generated Monte Carlo models with velocity perturbations of a few percent from an already well fitting starting model, we set up the starting models by hand and tested limits of the model space until the model still converged. In the statistics, we included only starting models which converged to a low chi². The two outlying models (now shown in Fig. 4c), "fast" and "slow", would lose their weight if we would add even more automated starting models in the centre of the model space. Thus, we think the statistics based on a few manually created starting models are supporting a robust final average model although based on a lower number of starting models.

The starting models used were 1D hanging below the seafloor along the profile now shown in Figure 4 as (c) and we added a sentence on the 1D structure of the starting models in the text.

• In addition, the authors may want to provide more details on this type of statistical test, right now is a bit vague. How are the initial models? Are they randomly created or they are derived from the forward modelling? The authors could add figures of the initial models, initial standard deviation, as well as the results from forward modelling in the supplementary material. Do they add Gaussian random noise to the picks (I would encourage them to add this to the test)?

Added now in 4c and as inlay in 4d to better document what the input for the modelling is. Random Gaussian noise was not added, but during modelling re-picking of phases (fine adjustments to the wavelet) did not lead to major differences in the resulting velocity model. We added a sentence:

*"Random Gaussian noise was not added, to the travel time picks, however, during modelling re-picking of phases (mainly fine adjustments to the picks) did not lead to major differences in the resulting velocity model."*

• The gravity modelling could be also improved as well. The authors could show how the gravity response derived from a density model with a homogeneous mantle density of 3.3 g/cc compares with the model they have and the observed anomaly. That would help to discern between serpentinized mantle and non-altered mantle rock, which in turn would allow to strengthen the hypothesis of the paper.

The anomalies in the density have been related to the anomalies in seismic velocities. However, the influence of these localised anomalies that are buried by several kilometres of sediments is minor for the general trend of the model fit to the satellite-derived gravity data. Based solely on gravity data, we cannot judge if the mantle is serpentinised in these patches.

• Line 350-351 and all section 5.4: "seafloor spreading and formation of oceanic crust was not initiated during the extension of the Ligurian Basin.". I would be more cautious here, it seems that the authors are saying that there is no oceanic crust in the whole Ligurian basin. Extension in this basin increases from north to south and as in the Tyrrhenian formation processes may significantly change from the north (region imaged in this study) to the south.

The Ligurian Basin is the NE part of the Liguro-Provencial basin. The studied profile covers the SW part of the Ligurian Basin. Along the profile, we do not observe oceanic crust. Of course, oceanic crust may still occur in the larger Liguro-Provencial basin, but we rule out that there is any oceanic crust towards the NE, within the Ligurian Basin.

We carefully discuss this and point out that the COT might be nearby and oceanic crust occurs towards the S and/or SW as observed by Gailler at al. (2009).

Minor changes:

• Line 18: augmented -> complemented

Changed.

• Line 22-23: "exhumation of sub-continental mantle which eventually became serpentinised". According to the models of mantle exhumation crustal faulting initiates the hydration of the mantle during rifting. Thus, serpentinization occurs before the exhumation. The authors should modify this sentence accordingly.

Re-phrased.

• Section 3.2 The GEOLOG recorder. This section is a bit out of place since this is not a technical paper and thus, it distracts the reader from the main point. I suggest moving this section to supplementary material and briefly mentioning the GEOLOG recording system in section 3.1.

We keep this section in its place, since we cannot refer to a technical paper describing the data logger. The title describes clear what the reader can expect by reading this section and can jump to the next section if it becomes too technically. We included some more technical specifications on the airgun system as well (as recommended by reviewer #2) and hence give more background information on the data acquisition parameters.

• Line 154: The gravimetric data (Fig. 5a) show a change approx. 20 km south of OBS208. Please add the numbering of OBS in Figure 5a.

It is 20 km south of OBS209. Additionally, we now also give the corresponding profile kilometre for this change in the text. We added the OBS numbers as shown in Figure 2.

• Figure 1: There is a bracket missing in Rollet et al. (2002), and it would be good to see the numbering of the OBS/H shown in Fig. 2 instead of OBS/H 201-208-215.

We changed the numbering according to figure 2. Corrected the bracket.

---

## Author Comment (AC6) · 28 Feb 2020

The "author-changes-tracked" manuscript contains changes applied to the manuscript for all three referee comments.

Please also note the supplement to this comment:
https://www.solid-earth-discuss.net/se-2019-187/se-2019-187-AC6-supplement.pdf
* * *

---

## Referee Report (RR1)

**Response to reviewers**

We sincerely thank the three reviewers for their fair and constructive reviews. We appreciate the feedback given on the manuscript and carefully incorporated all points risen. Please find blow our answers for each comment in green coloured text. We re-arranged the discussion sections. While most of the text remained unchanged, we added some paragraphs related to the review comments.

We also modified the title, which now better summarised our findings.

New title:

**Seismic evidence for failed rifting in the Ligurian Basin, Western Alpine Domain**

Detailed responses to the referees:

**Reviewer #1 (Laurent Jolivet)**

This paper is a welcome complement to the current knowledge of the nature of the crust in the Mediterranean back-arc basins. A debate has been active for many years on the nature of the crust and the existence or not of true oceanic crust. Recent investigations in the Tyrrhenian Sea suggest that a large part of what was previously interpreted as oceanic crust would in fact be exhumed continental mantle. This new paper addresses this question on the example of the Ligurian Basin, the north-easternmost part of the Liguro-Provençal Basin. If oceanic crust was so far supposed to be present in the center of the basin, it was always described a atypical with large volcanic intrusions instead of a well-organized mid-ocean ridge. This new contribution shows clearly that along the entire profile, no oceanic crust is present and that exhumed mantle is found in the southern part. Whether there is or not true oceanic crust further south will remain debated but, at least for the Ligurian portion, the debate should now be closed. The paper is well written and easy to read. Although I am not a specialist of seismics I could understand the methodology and the discussion and the whole makes a convincing manuscript.

The only suggestion I have to address to the authors is to open the discussion to the more general point of back-arc rifting in the Mediterranean context. The findings described here have important consequences in terms of rifting dynamics. Why does true oceanic crust does not emplace in this sort of back-arc environment is an important question.

Indeed, a discussion on the back-arc rifting in the broader Mediterranean context would be interesting and worthwhile. We feel that we can provide only limited input on the raised question regarding the lack of oceanic crust in other back-arc environments in the Mediterranean solely based on the seismic profile P02 presented here. In fact, it is not known whether other back-arc environments show a similar structure as the Ligurian Basin. However, the presented new findings for the Ligurian Basin are of general interest in themselves and merit a presentation in a focussed way.
We added a more general introducing section to chapter 5.4 to show possible driving limits to the opening of the Ligurian back-arc basin:
*"The opening of the Ligurian Basin in a back-arc position during late Oligocene and early Miocene that was driven by the south-east retreating Apennines-Calabria-Maghrebides subduction zone (e.g. Doglioni et al., 1997; Faccenna et al., 1997; Carminati et al. 1998; Rehault et al., 1984). The shift of active expansion from the Ligurian basin to the Tyrrhenian Sea is considered as a result of the Alpine collision that locked the Corsica-Sardinia drift towards the east and slab break-offs along the northern African margin and along the Apennines (Carminati et al., 1998). Thus, the opening of the Ligurian Basin was limited in time and space."*

**Reviewer #2 (Jean-Xavier Dessa)**

General comments

The manuscript presents us with a study of crustal structures over a profile crossing along strike in the central Ligurian Basin, by means of seismic refraction and seismic reflection. The main finding of the paper is that a transition from thinned continental crust to unroofed mantle covered by thick sediments is observed from northeast to southwest along the profile. In other words, no mantle-derived oceanic crust seems to be observed in the "oceanic" part of the basin. On these grounds alone, the paper represents a very interesting contribution and is definitely worthy of publication. I have some comments on the manuscript, that can be ranked as minor but that are numerous though. I think addressing them thoroughly would help improve clarity and consistency.

Specific comments and suggested corrections

The line numbering in the word template is different to the created PDF, however, we think that we easily could find the sections pointed out.

• Line 60: Dessa et al., 2011 is not a relevant citation as far as the Corsican margin is concerned. Rollet et al.'s paper and perhaps a few others should rather be cited here.

Indeed, we changed the reference to Contrucci et al., 2001 and Rollet et al., 2002.

• Lines 102-103: Some more details would be welcome on the seismic source and its tuning. How many airguns? What minimum and maximum volumes? What depth of immersion? What frequency range? Considering the level of technical details provided on the GEOLOG data logger for instance, there is room for a bit more information here.

We included more details on the airgun system itself:

"A total of 1079 shots were fired by an ~89-liter (5420 inch³) G-gun array, consisting of 2 sub-arrays. Each sub-array with a cluster of 2x8.5 litres (520 inch³), followed by a cluster in the middle of 2x6.2 litres (2x380 inch³, port) and 2x4.1 litres (2x250 inch³, starboard), and the third cluster again of 2x8.5 litres for both sub-arrays. The array with a string distance of 12 m was towed at 8 m below the sea-surface and 40 m behind the vessel. A shot interval of 60 s resulted in a shot distance of ~123 m. The guns were shot at ~190 bar providing a dominant frequency band of approximately 5-70 Hz."

• Line 104: Still on the technical side, we are told that the symmetry of the direct arrival was used as a criterion to refine station locations. It makes perfect sense but it would be interesting to explain how this is done practically (i.e., how the position is updated from an observed asymmetry). Either a few words or a reference could be provided.

We included more details (Lines 110-113):

"For this purpose, the direct arrival was picked and the deviation between computed and real travel times was minimised by adjusting the OBS's position along the profile. Dislocation off-line cannot be corrected with this method. For 2D traveltime modelling, the stations were projected on to the profile."

• Line 120: Technical again, and I might be unfamiliar with some recent developments, but I fail to see how an atomic clock may control the sampling rate of an autonomous sea-bottom instrument. Should we understand that in other contexts (such as in a lab...), the data logger would have this capability to be fitted with an atomic clock?

Indeed, the internal clock used in some of the recorders is a clock, which is controlled by the oscillation frequency of atoms (i.e. caesium), giving very high accuracy, compared to quartz oscillators. On the other hand atomic clocks consume much more power (probably double the rate) than a usual simple quartz oscillator, which is a disadvantage for long term deployments.

• Line 142: Nitpicking a bit, but Figure 3 has no a) and b) panels and yet, a reference is made here to Fig. 3a (although the sense of it is clear).

The labels "a)" and "b)" have been too small. We enlarged it for better visibility and included a panel c).

• Line 149: The meaning of a "longer" Pg phase is unclear to me. Should we understand that it is observed along a larger range of offsets? It might be rephrased.

Re-phrased: *"Simultaneously, when phase Ps3 disappears (from OBS208 towards the north (compare to OBS209 (Fig. 2c), where Ps3 only occurs on the southern branch), an additional refracted phase (Pg) (green picks in Fig. 2c-2d) occurs with an increasing range of offsets observed on the stations and becomes longer northwards."*

• Line 153-155: The text is a bit confused here. 1) Fig. 2b is invoked but does not correspond to a northern station, as the sentence seems to imply; 2) It is not clear to me that the critical distance is larger to the north (this may have a lot to do with the fact that labels along axes in Figure 2 are not readable, see my specific comment on this point below); 3) OBS208 is not labelled in any figure and the discussed change in gravimetric data is not even located below, but 20 km south if the text is to be believed. Why then not giving the actual location of said change in km along the profile rather than with respect to a remote OBS? On a sidenote, to the best of my understanding, this location would correspond to a distance of 50 km along the profile, where I was not able to identify any change in the free air anomaly. Are we talking about the decrease observed from 60 km onward? Confusing indeed …

We re-phrased this section in order to make it clearer to the reader:

1) Re-phrased the explanation of Ps3 (yellow phase, Figure 2) and then explain the slight changes from south to north.
2) We increased the axis labels on Figure 2 and adjusted a little bit the position of the phase labels. The difference in critical distance between the northern and the southern stations is very small (~5-10 km) and is maybe better to see in Fig. 2a.
3) OBS208 was wrong! Changed to OBS209. To follow the order of Figures we removed the link to Fig. 5. It was simply to give the reader a view to the second dataset to directly follow the changes, since they are very small. We do not want to change the order of figures, since the gravity modelling is based on the seismic results.

Yes, 60 km along the profile and onward towards the north, and it should be seen from OBS209. Indeed a profile KM will make the description much clearer. Added.

• Line 154: Same remark as for Line 142. No a) and b) panels in Figure 5. Note that this figure is referred to in the text before any reference is made to Figure 4. Normally, figures are numbered in the same order as that in which they are called in the text.

Figure 5 does not really have a panel (a) and (b) since for example the colour scale of the gravity model is at the top of the figure. We modified the text so that we now call the Figures in the correct order.

• Line 159: We are told that picking was made on hydrophone data rather than geophone. Is there a justification to this? One would arguably expect a better sensitivity of geophones. Is there any issue with data quality on geophones?

The data quality of the geophones is commonly controlled by the instrument coupling to the seafloor and thus varies largely between different study areas. Sometimes the geophone shows a similar or better S/N ratio than the hydrophone on our data set, also dependent on the offset range. While single instruments show a similar quality than the hydrophone, overall the hydrophone data was more robust for all stations along the profile. We included the following sentence on the data quality:

*"The overall quality of the hydrophone data was slightly better compared to the vertical geophone channel, however, the vertical component was used for picking to confirm…"*

• Line 161: I do not find it completely obvious that using multiples yields more information to confirm layer boundaries. Could the authors provide a citation or a bit of explanation to support this claim?

We inserted and explained in the text and gave a citation (Meléndez et al., 2014):

*"The vertical seismometer component was used for picking to confirm and to complement the picks observed on the hydrophone channel. In addition, multiples were picked when above the noise level (because of constructive interference) and where primary waves are below the noise level (Meléndez et al., 2014)."*

• Line 166: Some more details explaining how the set of additional starting models were generated would be most welcome. On what assumptions were they built? Do they span a large area in the model space? Etc. This might even warrant a figure if it can be made synthetic enough.

We added some text to clarify:

*"To test the model space and its limits, starting models, ranging from velocities between 1.8 km/s and 2.5 km/s at the seafloor with different velocity gradients, and ranging from 4.5 km/s to 7.5 km/s at 12-13.5 km depth to mimic the different types of crust, were manually created using RAYINVR (Zelt, 1999)."*

• Line 167: The use of 2 criterion begs the question of travel time uncertainties and how they were assessed. No information is given about that and that too would be most welcome. A 2 criterion of 1 only bears relevance if there is a rigorous and objective way to estimate uncertainties.

We included a sentence on the size of pick uncertainties that were assigned to the different picked phases (Lines 178-180):

*"The picks were assigned pick uncertainties ranging from 20 ms for clear near offset phases (Ps1), 30 ms for intermediate offsets (Ps2 and Pg), and up to 50-70 ms for picks at larger offset (Pn and PmP) taking into account the decreased resolution due to the increased wave length of the seismic signal and the decreased signal-noise-ratio."*

• Line 171 and 175: Standard deviation values are provided in s here, which does not make any sense ito me and is not coherent with Figure 4b an 4c, where they are given in km/s, as one would expect for a velocity model.

Corrected.

• Line 181-182: To back up the claim that lateral velocity variations are a consequence of the irregularity of the salt layer, it would be interesting to compare the wavelengths of these anomalies with those of the salt unit as imaged in MCS data. This would furthermore provide some added value to the MCS data which are practically of no use in the discussion of structures as it stands (let alone parasound data)—this point is discussed below (comment on Fig. 3).

We inserted a third panel (Fig. 3b) which compares the MCS data with the OBS data within the range of 300 traces and linked this figure to the text.

• Line 199: Error of reference: the histogram is not in Figure 5a (which does not exist), but in Figure 4e.

Changed.

• Line 217: The title of the section is wrong. "Discussion" rather than "Introduction" should feature here. The section title is likely inherited from the manuscript template...

Corrected - wrong copy paste into the template.

• Line 224: Same figure, same problem as in Line 199. Figure 5e instead of 4e.

Corrected.

• Line 267-268: It is not clear to me what the authors mean with a less evolved crust. Do they mean the thickness of it? Its nature? As a result, I find the meaning of the last two sentences of this paragraph rather enigmatic.

Changed "less evolved" to "less thick". Added that these observations indicate thickening continental crust towards the North.

• Line 309: I find the sentence about the preference of the authors for a magmatic origin to the observed magnetic anomalies rather than a relation with an unmapped spreading axis to be quite an understatement as their own results seem to completely rule out the possibility of a ridge axis (no oceanic crust is an overwhelming argument against the existence of any accretion axis at any time here I believe). More generally, I think this very interesting result and its implications are not highlighted enough in the discussion.

We changed the title in order to focus on our main finding. We changed the abstract and the style of the conclusion to bullet points to improve the visibility of our findings and highlight them.

• Fig. 1: A lot of features on the map, some not very visible, some practically invisible due to a poor choice of non-contrasting colors with respect to the bathymetric background or to the use of thin lines. Rollet et al. refer to "atypical oceanic domain" instead of "atypical oceanic crust". Since the main result of this study is to rule out the existence of an oceanic igneous crust, the reference to an "Atypical Oceanic Crust" is a bit confusing here. I would suggest the same appellation as in Rollet et al.

Changed to atypical oceanic domain (AOD) in the figure. Reduced the contrast of the map and enlarged the contrast and thickness of the lines and objects of importance.

• Fig. 2: As mentioned above, labels along axes should be made readable for all data plots.

Axis labels enlarged and phase labels slightly adjusted.

• Fig. 3: Features in the MCS profile are barely visible and poorly discussed in the text. This observation holds even more true for parasound results which do not back results at all. I think dropping them could be considered.

We inserted a third part into the figure (Fig 3, panel b) that compares the undulations of the first arrival phase in the OBS with the MCS data. We like to keep this figure, since it provides a good impression on the complexity of the shallow portion of the subsurface and shows the entire data range acquired along this profile.

**Reviewer #3 (Manel Prada)**

General comments

• The manuscript "Oligocene-Miocene extension led to mantle exhumation in the central Ligurian Basin, Western Alpine Domain" by A. Dannowski and co-authors present new constraints on the petrological nature of the basement in the Ligurian Basin from new wide-angle seismic data and travel-time tomography. The authors show that rather than oceanic crust, as previously interpreted in the area, the northwestern region of the basin experienced crustal thinning and later mantle exhumation. However, I found the occurrence of mantle exhumation difficult to reconcile with the velocity structure of the uppermost mantle presented here. Considering that the mantle is fully exposed to the seawater during the opening of the basins, I found strange that the top of the mantle does not show the typical velocity gradient of exhumed mantle regions, in which Vp increases progressively from ~4.6 km/s (100 % alteration) to 7.8-8.0 km/s (no alteration) (Minshull 2009; https://doi.org/10.1016/j.crte.2008.09.003; Prada et al., 2015 doi: 10.1093/gji/ggv271). While I agree that there is no oceanic crust, the lack of an exhumed mantle-like Vp vertical gradient implies that the mantle was not fully hydrated and thus, exhumed, possibly because of the presence of syn-rift sediments or the existence of hyperextended continental crust. In fact, this interpretation fits nicely with the model. Lower crustal velocities are > 5.5 km/s, which may well be indicative of tilted fault blocks and rotated syn-rift sediments (e.g. Bayracki et al., 2016, Nature Geoscience, DOI: 10.1038/NGEO2671). The top of the continental basement in these settings can be really rough, and thus difficult to identify in OBS data. The fact that you don't see it, doesn't mean it's not there. In addition, mantle Vp is close to 8 km/s in some regions (e.g. beneath OBS205), while it decreases in others to < 7.5 km/s. This pattern resembles the mantle structure underlying continental tilted blocks reported in other rifted margins such as Galicia (Bayracki et al., 2016) and the Porcupine Basin (Prada et al., 2017 EPSL; http://dx.doi.org/10.1016/j.epsl.2017.06.040). Such pattern is attributed to the fault-controlled water influx to the mantle during rifting (Bayracki et al., 2016). In light of these observations, I advise the authors to reconsider their interpretation. Apart from this aspect, I also found some issues during the modelling and in section 5.4 that, if tackled, can help to improve the robustness of the final model, and thus, strengthen the paper. I discuss them the bellow.

We agree that the concept of mantle exhumation poses some interesting aspects and thank the reviewer for pointing these out. Indeed our data may not provide the information if sediments are underlain by thinned continental crust or exhumed mantle. We now discuss both scenarios. Exhumed mantle does not necessarily imply to be exposed directly on the seafloor as we mentioned in lines 270-275. We now additionally clarify this in the discussion by adding extra paragraphs in sections 5.1 and 5.2. We re-phrased the manuscript at several places to open up the discussion about the continental material (thinned continental crust or exhumed subcontinental mantle) underlying Ligurian Basin.

In contrast to the Tyrrhenian Sea we observe a strong in amplitude PmP reflection, which indicates a high velocity contrast at the crust-mantle boundary. The nature of the velocities >5.5 km/s can be debated and we cannot distinguish between fast sediments and left over rotated continental crust blocks (rotated the orthogonal direction to the profile). Sediments have to play a role during rifting and mantle serpentinisation (Ruepke et al., 2013), else we would observe a much higher rate of mantle serpentinisation, we agree, also in areas with remnant blocks of continental crust.

We changed the manuscript title to: *"Seismic evidence for failed rifting in the Ligurian Basin, Western Alpine Domain"*

Regardless of these issues, the paper unequivocally demonstrates that there is no oceanic crust in this region of the Ligurian Basin, and that is of great relevance for the community working on the

Mediterranean region. This study fits nicely with the goals of Solid Earth, and thus, I strongly encourage the authors to tackle all these aspects and resubmit the manuscript for its publication.
Best regards, Manel Prada
Major issues:

• The authors use forward modelling, I presume, to explore the lateral consistency of the seismic phases observed in each receiver. Then, they use this preliminary model as input for the tomography. However, it is confusing the way the authors describe and apply the layer stripping strategy. The authors say, "In a first step only near offset picks with distances smaller than 15 km were inverted." This is rather confusing. It seems that the authors have inverted the travel times within 15 km of offset from each receiver, independently of the seismic phase they correspond to. It would be better to explicitly mention the type of seismic phases that the authors have included in the first step, which I guess by Figure 4, are all sedimentary and crustal phases, plus PmP.

We removed the sentence since it is rather confusing and not necessary to explain the modelling strategy for the final average velocity model. This actually was a process that allowed us to get better acquainted with the data and the model behaviour and which will not be visible in the final results. But indeed, in first steps we only selected picks up to 15 km offset to image the uppermost sediments.

• One would also appreciate more details on the layer stripping strategy. Did the authors overdamped the result of the crustal layer when inverting for mantle phases?

We overdamped the model and included this in the text.

• On the other hand, the authors follow some sort of Monte Carlo analysis to assess the space of possible solutions but they only use 17 models for the crustal level and even a lower number for mantle phases, 12. The final standard deviation is low in Fig. 4. My concern is that given the low number of realizations tested the initial standard deviation (which one would appreciate seeing in the supplementary material) might below as well. I suggest testing at least 100 models for each layer, which is what is commonly done in this type of study to assess the uncertainty of model parameters. The outcome of this uncertainty analysis in its present form is not convincing which may lead to skepticism of the final interpretation.

In contrast to 100 or even 1000 different models, to call it Monte Carlo analysis, we preferred to calculate a smaller set of models to test a wider model space. In contrast to the commonly performed automated generated Monte Carlo models with velocity perturbations of a few percent from an already well fitting starting model, we set up the starting models by hand and tested limits of the model space until the model still converged. In the statistics, we included only starting models which converged to a low chi². The two outlying models (now shown in Fig. 4c), "fast" and "slow", would lose their weight if we would add even more automated starting models in the centre of the model space. Thus, we think the statistics based on a few manually created starting models are supporting a robust final average model although based on a lower number of starting models.

The starting models used were 1D hanging below the seafloor along the profile now shown in Figure 4 as (c) and we added a sentence on the 1D structure of the starting models in the text.

• In addition, the authors may want to provide more details on this type of statistical test, right now is a bit vague. How are the initial models? Are they randomly created or they are derived from the forward modelling? The authors could add figures of the initial models, initial standard deviation, as well as the results from forward modelling in the supplementary material. Do they add Gaussian random noise to the picks (I would encourage them to add this to the test)?

Added now in 4c and as inlay in 4d to better document what the input for the modelling is. Random Gaussian noise was not added, but during modelling re-picking of phases (fine adjustments to the wavelet) did not lead to major differences in the resulting velocity model. We added a sentence:

*"Random Gaussian noise was not added, to the travel time picks, however, during modelling re-picking of phases (mainly fine adjustments to the picks) did not lead to major differences in the resulting velocity model."*

• The gravity modelling could be also improved as well. The authors could show how the gravity response derived from a density model with a homogeneous mantle density of 3.3 g/cc compares with the model they have and the observed anomaly. That would help to discern between serpentinized mantle and non-altered mantle rock, which in turn would allow to strengthen the hypothesis of the paper.

The anomalies in the density have been related to the anomalies in seismic velocities. However, the influence of these localised anomalies that are buried by several kilometres of sediments is minor for the general trend of the model fit to the satellite-derived gravity data. Based solely on gravity data, we cannot judge if the mantle is serpentinised in these patches.

• Line 350-351 and all section 5.4: "seafloor spreading and formation of oceanic crust was not initiated during the extension of the Ligurian Basin.". I would be more cautious here, it seems that the authors are saying that there is no oceanic crust in the whole Ligurian basin. Extension in this basin increases from north to south and as in the Tyrrhenian formation processes may significantly change from the north (region imaged in this study) to the south.

The Ligurian Basin is the NE part of the Liguro-Provencial basin. The studied profile covers the SW part of the Ligurian Basin. Along the profile, we do not observe oceanic crust. Of course, oceanic crust may still occur in the larger Liguro-Provencial basin, but we rule out that there is any oceanic crust towards the NE, within the Ligurian Basin.

We carefully discuss this and point out that the COT might be nearby and oceanic crust occurs towards the S and/or SW as observed by Gailler at al. (2009).

Minor changes:

• Line 18: augmented -> complemented

Changed.

• Line 22-23: "exhumation of sub-continental mantle which eventually became serpentinised". According to the models of mantle exhumation crustal faulting initiates the hydration of the mantle during rifting. Thus, serpentinization occurs before the exhumation. The authors should modify this sentence accordingly.

Re-phrased.

• Section 3.2 The GEOLOG recorder. This section is a bit out of place since this is not a technical paper and thus, it distracts the reader from the main point. I suggest moving this section to supplementary material and briefly mentioning the GEOLOG recording system in section 3.1.

We keep this section in its place, since we cannot refer to a technical paper describing the data logger. The title describes clear what the reader can expect by reading this section and can jump to the next section if it becomes too technically. We included some more technical specifications on the airgun system as well (as recommended by reviewer #2) and hence give more background information on the data acquisition parameters.

• Line 154: The gravimetric data (Fig. 5a) show a change approx. 20 km south of OBS208. Please add the numbering of OBS in Figure 5a.

It is 20 km south of OBS209. Additionally, we now also give the corresponding profile kilometre for this change in the text. We added the OBS numbers as shown in Figure 2.

• Figure 1: There is a bracket missing in Rollet et al. (2002), and it would be good to see the numbering of the OBS/H shown in Fig. 2 instead of OBS/H 201-208-215.

We changed the numbering according to figure 2. Corrected the bracket.

[revised manuscript text omitted]

---

## Author Response (AR2)

Dear editor and reviewer,

we sincerely thank Manel Prada for reading through the revised manuscript and point out some issues that need more clarification. We carefully worked through all the raised points. Please find below our answers for each comment in green coloured text. We used the revised manuscript to track the new changes. Line numbers given in our response correspond to

5   the manuscript without "track changes" visible.

On behalf of the authors,

with kind regards,

Anke Dannowski

10  **Reviewer report (Manel Prada)**

The authors have addressed most of the issues raised in the previous review, and the paper has improved considerably. However, as I mention in the attached PDF, there are minor to moderate issues that need to be explained/addressed to make the paper consistent.

The main moderate issue is that the authors should be more consistent with their main interpretation regarding the presence of

15  exhumed mantle (i.e. whether Oligocene-Miocene extension led to extreme thinned continental crust or exhumed subcontinental mantle with a low grade of mantle serpentinisation remains enigmatic.) There are certain sections where the authors seem to favor the full exhumation of the mantle as opposed to the presence of hyperextended crust, which from the data they provide is not possible to discern.

Apart from this, I proposed some minor changes to the text and figures in the attached PDF file.

**Point by point reply:**

Line 20 - perhaps the authors could mention here their hypothesis for the low grade of serpentinzation (i.e. high sedimentation rate during syn-rift stage).

We included the hypothesis and re-phrased this sentence to: *"... thinned continental crust or exhumed subcontinental mantle*

25  *remains enigmatic. A low grade of mantle serpentinisation indicates a high rate of syn-rift sedimentation."*

Line 23 - this second sentence is a bit redundant with the first part of the sentence, please remove.

Deleted the second part of the sentence and re-phrased the two sentences to: "*However, rifting failed before oceanic spreading was initiated and continental crust thickens towards the NE within the northern Ligurian Basin.*"

Line 23 – Change "*is thickening*" to "*thickens*"

30  Changed

Line 45 – Change "*seismic refraction line*" to "*seismic refraction data*"

Changed

Line 182 – Change "*The starting models used in the analysis were 1D hanging below the seafloor (Fig. 4c).*" to "*The 1D starting models used were hanging below the seafloor (Fig. 4c).*".

35    Changed

197 – Change "*crust, i.e. the crystalline basement.*" to "*(i.e. crystalline basement; CB in Fig. 4a)*"

Changed to "*crust, i.e. the crystalline basement (CB in Fig. 4a).*"

Line 211 – "*We interpret this section as syn-rift sediments, possibly …*"

To be consistent with the abstract and the uncertainty of the nature of the rock at these depths the authors should also mention

40    the possibility here that these velocities account for tilted fault blocks of continental curst.

We added the sentence: "*We will discuss the nature of these layers in a later section, since the observed seismic velocities also account for tilted fault blocks of stretched continental crust.*"

Line 244 – Change the second part of the sentence from "*that no typical oceanic crust and no thick layer of gabbroic rocks is present along the profile*" to "*the lack of a thick gabbroic layer, and thus, the lack of typical oceanic crust*".

45    Changed.

246 – Change "*in the Tyrrhenian Sea (Prada …).*" to "*in the Central Tyrrhenian Sea (Prada …).*".

Changed

Line 253 - As the authors mention later in this section (L. 282-284), they do not know weather exhumation or hyper-extension of the crust occurs in this region. I would suggest to remove this sentence to be consistent with the message of this section.

50    We would like to keep the second part of the sentence, since by this we open the discussion in the next section.

258 – Change "*high uncertainty*" to "*higher uncertainty*"

Changed

Line 270 – Change "*A continental crustal thickening*" to "*The thickening of the continental crust*".

Changed

55    Line 271 – Change "modelling of the free air anomaly (Fig. 5)." to "gravity modelling (Fig. 5)".

Changed

Lines 271 to 274 - This paragraph is not clear, and does not provide much as you already mention the lack of oceanic crust. I suggest to remove it.

We like to keep this sentence, because it supports our discussion that we observe continental crust. The observation of crustal

60    thickening towards the NE contradicts an oceanic type of crust which we would expect to thin out towards the NE. We changed the last part of the sentence to make it more clearly to: "*… an abrupt change from oceanic to continental crust or oceanic crust to gradually thin out towards the NE.*".

Line 285 – Delete "very well"

Deleted

65    Line 285 - I'm confused here. The authors interpret Vp  between 4.5-5.7 km/s as syn-rift sediments (L. 224-225), then they suggest that mantle exhumation may have occurred (L. 267), but now it seems that their interpretation is unequivocally that hyper-extended crust is overlying the mantle. The authors need to be consistent along the manuscript and support a unique

interpretation (which may include both syn-rift or hyper-extended crust as they have way to discern between them), otherwise it turns to be rather confusing.

70 We try to clarify this and re-phrased this sentence at the end. This comparison is related to the northern end of our profile, where we indeed observe continental crust velocities. We changed this paragraph to: "*Our profile only provides information on the basin centre where the absolute velocities of Le Douaran et al. (1984) and Contrucci et al. (2001) fit continental crust velocities and support our interpretation for the northern end of the profile where we observe mantle material beneath thinned continental crust.*"

75 Line 289 – Remove the sentence: "*Seismic velocities of ~7.3 km/s and higher are too fast for magmatic crust (Grevemeyer et al., 2018; Christeson et al., 2019).*".

Removed

Line 296 – Remove "*or a result of mafic intrusion*" - a mafic (e.g. gabbro) intrusion cannot have Vp>7.8 km/s, perhaps ultramafic rocks may reach to that velocities, and in this setting there is no evidence of crustal magmatic velocities as the

80 authors say, so that I would remove this.

Removed

Lines 296 to 299 – Change "*that syn-rift sediments (nowadays showing high P-wave velocities) may have been directly deposited on top of the mantle or brittle continental crust (Pérez-Gussinyé, 2013).*" to "*that either contiental crustal blocks are overlying the mantle, and thus it was not fully exposed during rifting or that syn-rift sediments (nowadays showing high*

85 *P-wave velocities) may have been directly deposited on top of the mantle preventing its full serpentinisation (Perez-Gussinyé et al., 2013).*".

Changed

Line 300 – This is and overstatement, as there are regions of the Atlantic margin that may not be as serpentinized as the Goban Spur and the Iberia margin. So perhaps the best here is to keep the comparison to these two regions rather than the entire

90 Atlantic margin. In addition I would encourage the authors to include some values of serpentinisation from the Goban Spur and Iberia, so the reader can have a sense on how much different is the degree of serpentinzation between the presented Atlantic margins and the Ligurian.

We re-phrased the sentence to "*However, the fast mantle in the Ligurian Sea would support a much lower degree of serpentinisation when compared to these two regions with a mantle serpentinisation of 100% at the top basement and <25%*

95 *in 5-7 km depth at the Goban Spur and >75% at the top basement and <25% in 2 km depth at the Iberia margin.*"

Line 304 – Change "*In comparison to the Tyrrhenian Sea, …*" to "*In comparison to the Central Tyrrhenian Sea, where exhumed mantle is inferred, …*"

Changed

Line 305 – Change "*high (Fig. 4a).*" to "*faster (Fig. 4a).*"

100 Changed

Line 305 – Change "*Tyrrhenian Sea Domain #3*" to "*Magnaghi and Vavilov basins in the Tyrrhenain Sea (i.e. Domain #3 in Prada et al., 2014)*".

Changed

Lines 310 to 314 – The Ligurian has a thicker syn (?) and post-rift layer of sediments than that of Domain #3 in the Tyrrhenian, that is a fact I agree. However, there is no quantitative evidence (at least is not provided here) that the Ligurian had a higher sedimentation rate than the Tyrrhenian.

The Ligurian is far older than the Domain #3 in the Tyrrhenian, which did not formed 9-10 Ma ago as the authors refer, but <5.9 Ma ago (Kastens and Mascle, et al., 1990 ODP results; Prada et al., 2016) and, consequently really fast (> 2-3 cm/yr ; Faccena et al., 2001). Thus, the Ligurian stopped opening 10Ma earlier than the opening of Domain #3 at 15-16Ma. 10 Ma gives a lot of time for sediments to cover up the basin, so there is no need to invoke a higher sediment rate to explain the difference in sediment thickness, but the difference in age of formation.

The authors use the presence of syn-rift sediments from the Gult of Lions to reinforce the hypothesis that a thicker syn-rift cover might be overlying the mantle in the Ligurian than in the Tyrrhenian. This could be a possibility, but the authors need to consider also the fact that the continental margin and thus the source of syn-rift sedimentation was closer to the center of the basin during the formation of the Ligurian than during the formation of Domain #3 in the Tyrrhenian, given that the Sardinia and Campanian margins were already formed, thus the center of the basin (where Domain #3 formed) in the Tyrrhenian was already at least 100 km away from emerged land (Prada et al., 2014).

Overall, once could would agree that perhaps the syn-rift sedimentary cover is thicker in the Ligurian than in Domain #3 in the Tyrrhenian because of the latter aspect (i.e. proximity of the margin to the center of the basin). In contrast, the hypothesis of a higher sedimentation rate looks rather weak and not supported by any quantitative observation.

We agree entirely with the reviewer's comments and we re-phrased this section to make it more clear to the reader. We corrected the timing of the opening of the Tyrrhenian Sea and divided it to extension and spreading and provided the references. We changed the order of the observed differences between both basins and discuss them separately. While we used most of the phrasings we changed the order and added/changed 2 sentences. These changes are summarised in the one changed section:

"*(2) The Ligurian Basin was stretched ~150 km during the ~16 million year opening phase, while the Tyrrhenian Sea was stretched ~170 km within ~5 million years prior the onset of oceanic spreading (Faccenna et al., 2001, Prada et al., 2014); thus, the stretching rate in the Tyrrhenian Sea was higher compared to the Ligurian Sea. Further, (3) the Ligurian Basin has a thick sedimentary cover of ~6-8 km, while the Tyrrhenian Sea Domain #3 shows a sedimentary cover of ~1-2 km (Prada et al., 2014). Syn-rift sedimentation was recorded in MCS data (Fig. 1, inlay profile 3) in the Gulf of Lion (Jolivet et al., 2015). The proximity of the two basins to the continental margin during their formation might result in a different syn-rift sedimentation rate that possibly was higher in the Ligurian Sea compared to the Tyrrhenian Sea.*"

Line 320 – This sentence needs to be rephrased, right now it does not make much sense.

*Re-phrased to: "Also the interpretation of extremely thinned brittle continental crust requires syn-rift sedimentation since the stretching might open fluid pathways through the crust down to the mantle and would lead to a high degree of mantle serpentinisation (Nagel and Buck, 2004)."*

Line 336 – Change "The exhumation of" to "*The nature of the COT along the Gulf of Lions is still debated*".

*We ex-changed the entire sentence to: "The nature of the COT along the Gulf of Lion is still debated.".*

Line 343 – Change "… *imaged in the travel time tomographic approach with the typical pattern observed at mid-ocean ridges (Gailler et al., 2009), with a high velocity gradient in the upper oceanic crust and a low velocity gradient in the lower crust.*"

to "… *interpreted on the basis of a 2D Vp model derived from travel time tomography as an anomalously thin oceanic crust (4–5 km) with the typical two-layer gradients clearly characteristic of oceanic layers 2 and 3 (Gailler et al. 2009).*"

*Changed to: "… interpreted on the basis of a 2D P-wave model derived from travel time tomography as an anomalously thin oceanic crust (4–5 km) with the typical two-layer gradients clearly characteristic of oceanic Layers 2 and 3 (Gailler et al., 2009)."*

Line 362 – Change "…*,which was indeed not mapped in MCS data so far.*" to "…*, absent in MCS data so far.*"

*We like to keep the original phrasing.*

Line 365 - careful here, calc-alkaline signature is typical from subduction-related melts, not with post-rift magmatism. The fact that makes this seamount a potential post-rift feature is the age of basalts, which are younger than the age of the rifting (16Ma). THe signature of the melt can be biased by the contamination of the subduction-related fluids.

*Change to: "… 2012). The age is clearly indicating …".*

Line 390 – Yet this phase is not certain from your observations. The authors should emphasize on that.

*We included the sentence: "Yet, it is debated if we observe the latter phase in the central Ligurian Basin.".*

Line 403 – Delete the sentence and add the information to the previous one: *"... pattern of magmatic dmains, supporting again the lack of oceanic spreading during the formation of the Ligurian basin in the Oligocene-Miocene.*"

*Changed*

Line 403 to 405

This is confusing towards the North where the Makris line is the crust thickens, thus the mantle won't exhume because there is not enough extension, not because it's not lasting enough. I would remove this sentence.

*Changed to: "Continuing further north of our seismic line, the Oligocene-Miocene extension led to thinned continental crust, but the amount of extension was too small to extremely thin out the continental crust and exhume mantle."*

Line 412 – add "*syn-rift*" and change "*thinned*" to "*thin*".

*Changed*

Line 416 - Again, from your data it is hard to conclude that the mantle is fully exposed. The authors should leave it as a they state above:

"*it remains enigmatic if the mantle is overlain directly by sediments or by extremely thinned continental crust of up to 2.5 km*"

We added this sentence: *"It remains enigmatic if the mantle is overlain directly by sediments or by extremely thinned continental crust of up to 2.5 km thickness."*

Line 424 – Delete "*however, nearby*"

Removed

170 Figure 1 – yellow and orange triangles are mentioned in the figure caption 3 but not in this one. It should included here as it is the first figure where they appear.

Added

Figure 4 – the inset showing the 1D profiles in (d) needs to be larger as (c).

We keep it as it is, since the scale of the 1D profiles in Fig. 4d is horizontally the same and vertically larger than in 4c. It is an 175 important background information on the modelling, however, we feel it is large enough to spot the tested model space.

**Marked-up manuscript:**

[revised manuscript text omitted]

---

## Author Response (AR3)

Dear Editors,

on behalf of my co-authors I thank you very much for accepting our manuscript and for the smooth and professional review processes. Special thanks goes to the reviewers that motivated us to go back to details and that gave us large support to improve the manuscript.

We will recommend the journal to others and will also consider it for further publications. I think it is great to have the review process open and as a kind of discussion.

Kind regards,
Anke Dannowski